# Diverse evolutionary pathways challenge the use of collateral sensitivity as a strategy to suppress resistance

**Rebecca EK Mandt[1]\*, Madeline R Luth[2], Mark A Tye[3,4], Ralph Mazitschek[1,3], Sabine Ottilie[2], Elizabeth A Winzeler[2,5], Maria Jose Lafuente-Monasterio[6], Francisco Javier Gamo[6], Dyann F Wirth[1,7], Amanda K Lukens[1,7]\***

[1]Department of Immunology and Infectious Diseases, Harvard T.H. Chan School of Public Health, Boston, United States; [2]Division of Host Pathogen Systems and Therapeutics, Department of Pediatrics, University of California, San Diego, San Diego, United States; [3]Center for Systems Biology, Massachusetts General Hospital, Boston, United States; [4]Harvard Graduate School of Arts and Sciences, Cambridge, United States; [5]Skaggs School of Pharmaceutical Sciences, University of California, San Diego, La Jolla, United States; [6]Tres Cantos Medicines Development Campus, Diseases of the Developing World, GlaxoSmithKline, Madrid, Spain; [7]Infectious Disease and Microbiome Program, The Broad Institute, Cambridge, United States

**\*For correspondence:**
rebeccamandt@gmail.com
(REKM);
alukens@broadinstitute.org
(AKL)

**Abstract** Drug resistance remains a major obstacle to malaria control and eradication efforts, necessitating the development of novel therapeutic strategies to treat this disease. Drug combinations based on collateral sensitivity, wherein resistance to one drug causes increased sensitivity to the partner drug, have been proposed as an evolutionary strategy to suppress the emergence of resistance in pathogen populations. In this study, we explore collateral sensitivity between compounds targeting the *Plasmodium* dihydroorotate dehydrogenase (DHODH). We profiled the cross-resistance and collateral sensitivity phenotypes of several DHODH mutant lines to a diverse panel of DHODH inhibitors. We focus on one compound, TCMDC-125334, which was active against all mutant lines tested, including the DHODH C276Y line, which arose in selections with the clinical candidate DSM265. In six selections with TCMDC-125334, the most common mechanism of resistance to this compound was copy number variation of the *dhodh* locus, although we did identify one mutation, DHODH I263S, which conferred resistance to TCMDC-125334 but not DSM265. We found that selection of the DHODH C276Y mutant with TCMDC-125334 yielded additional genetic changes in the *dhodh* locus. These double mutant parasites exhibited decreased sensitivity to TCMDC-125334 and were highly resistant to DSM265. Finally, we tested whether collateral sensitivity could be exploited to suppress the emergence of resistance in the context of combination treatment by exposing wildtype parasites to both DSM265 and TCMDC-125334 simultaneously. This selected for parasites with a DHODH V532A mutation which were cross-resistant to both compounds and were as fit as the wildtype parent in vitro. The emergence of these cross-resistant, evolutionarily fit parasites highlights the mutational flexibility of the DHODH enzyme.

## Editor's evaluation

This study addresses an important question in the field of antimicrobial chemotherapy: whether combinations of enzyme inhibitors that select for mutations that confer resistance to one inhibitor and at the same time increased sensitization to the other inhibitor, can provide a path towards mitigating resistance risks. The authors investigated one such combination of inhibitors of *Plasmodium*

*falciparum* DHODH (dihydroorotate dehydrogenase), finding that despite "collateral sensitivity", it was still possible to select parasites with resistance to both inhibitors without any change in parasite fitness. Additional cross-susceptibility and structural modelling strengthen this study, which is performed to a high technical standard and presents a convincing body of data.

## Introduction

Antimicrobial resistance threatens our ability to combat multiple infectious agents, and is widely recognized as one of the greatest public health threats of the 21st century (*Jee et al., 2018*). Globally, drug resistance is a recurring obstacle in efforts to control many endemic infectious diseases, and malaria is no exception. The emergence of resistance in *Plasmodium* parasite populations and subsequent treatment failure has been documented for all antimalarial drugs in clinical use, including frontline artemisinin combination therapies (*Haldar et al., 2018*). Thus, there is an ongoing need to develop a next generation of drugs targeting distinct vulnerabilities of *Plasmodium* parasites.

Additionally, understanding the pathways and susceptibility to drug resistance is important to assess next-generation antimalarials, ideally while they are still early in the development process. Inhibitors targeting the *Plasmodium* dihydroorotate dehydrogenase (DHODH) offer an illustrative example. DHODH is an electron transport chain protein that catalyzes the oxidation of dihydroorotate in orotic acid, which is the rate-limiting step of pyrimidine biosynthesis. In *Plasmodium,* this reaction is dependent on flavin mononucleotide and ubiquinone cofactors and is coupled to mitochondrial respiration. Because *Plasmodium* lacks pyrimidine scavenging pathways, this enzyme is essential for parasite growth and survival (*Phillips and Rathod, 2010*). Multiple high-throughput screens have identified DHODH as a drug target in *Plasmodium falciparum* (*Baldwin et al., 2005*; *Pavadai et al., 2016*; *Patel et al., 2008*; *Ross et al., 2018*; *Phillips et al., 2008*), and the triazolopyrimidine-based DHODH inhibitor DSM265 was developed as a clinical antimalarial candidate (*Phillips et al., 2015*).

While the clinical DHODH inhibitor candidate DSM265 showed promising activity in pre-clinical and clinical studies (*Llanos-Cuentas et al., 2018*; *McCarthy et al., 2017*; *Murphy et al., 2018*; *Phillips et al., 2015*; *Sulyok et al., 2017*), we previously demonstrated that resistance to DSM265 emerges rapidly both in vitro and in a humanized mouse model of *P. falciparum* infection (*Mandt et al., 2019*). Resistance was primarily conferred by point mutations in the *dhodh* locus. In Phase 2 clinical trials with DSM265, 2 out of 24 patients failed treatment due to resistance, with recrudescent parasites harboring mutations in *dhodh* including DHODH C276Y, C276F, and G181S (*Llanos-Cuentas et al., 2018*). The C276Y and C276F mutations were also observed in in vitro resistance selections, as was DHODH G181C (*Phillips et al., 2015*; *Mandt et al., 2019*; *White et al., 2019*). The point mutations C276F and G181D were observed in our in vivo selection model (*Mandt et al., 2019*).

A common strategy to counter the evolution of resistance is to utilize a combination of two or more drugs with distinct antimicrobial mechanisms. The basic rationale for this strategy is that even if there is some probability of an organism acquiring resistance to each drug separately, the probability that two resistance mutations would occur simultaneously is much lower. Combination therapy is the standard of treatment for malaria (*Hastings, 2011*), tuberculosis (*Kerantzas and Jacobs, 2017*), and HIV/AIDS (*Cihlar and Fordyce, 2016*), and is increasingly being recommended for the treatment of other bacterial infections (*Schmid et al., 2019*; *Coates et al., 2020*; *Bodie et al., 2019*). However, multidrug resistance has emerged in many major human pathogens, complicating patient treatment and threatening public health efforts to control the spread of disease (*Hamilton et al., 2019*; *Manson et al., 2017*; *Dheda et al., 2017*; *Gregson et al., 2017*). Large-scale bacterial evolution studies have also shown that certain combinations of inhibitors can actually accelerate the emergence of resistance compared to treatment with a single inhibitor (*Hegreness et al., 2008*; *Dean et al., 2020*). Thus, there is a need to re-evaluate the current treatment approach, and to strategically design combination therapies based on their ability to slow or suppress the emergence of resistance.

One possible strategy takes advantage of collateral sensitivity, in which resistance to one drug causes increased sensitivity to another (*Baym et al., 2016*). The phenomenon of collateral sensitivity, or 'negative cross-resistance', has been observed across various antibiotic classes in bacteria, as well as anti-fungals, and even cancer therapies (*Imamovic and Sommer, 2013*; *Rosenkilde et al., 2019*; *Maltas and Wood, 2019*; *Rank et al., 1975*; *Leroux et al., 2000*; *Dhawan et al., 2017*; *Lorendeau et al., 2017*). Laboratory studies evaluating the evolution of resistance to antibiotics have demonstrated

**eLife digest** Malaria affects around 240 million people around the world every year. The microscopic parasite responsible for the disease are carried by certain mosquitoes and gets transmitted to humans through bites. These parasites are increasingly acquiring genetic mutations that make anti-malaria medication less effective, creating an urgent need for alternative treatment approaches.

Several new malaria drugs being explored in preclinical research work by binding to an enzyme known as DHODH and preventing it from performing its usual role in the parasite. Previous work found that, in some cases, malaria parasites that evolved resistance to one type of DHODH inhibitor (by acquiring mutations in their DHODH enzyme) then became more vulnerable to another kind. It may be possible to leverage this 'collateral sensitivity' by designing treatments which combine two DHODH inhibitors and therefore make it harder for the parasites to evolve resistance.

To investigate this possibility, Mandt et al. first tested several DHODH inhibitors to find the one that was most potent against drug-resistant parasites. In subsequent experiments, they combined TCMDC-125334, the best candidate that emerged from these tests, with a DHODH inhibitor that works well against vulnerable parasites. However, the parasites still rapidly evolved resistance. Further work identified a new DHODH mutation that allowed the parasites to evade both drugs simultaneously.

Together, these findings suggest that the DHODH enzyme may not be the best target for new malaria drugs because many it can acquire many possible mutations that confer resistance. Such results may inform other studies that aim to harness collateral sensitivity to fight against a range of harmful agents.

that when resistance to one inhibitor leads to collateral sensitivity to another, the two inhibitors can be used in a sequential cycling strategy to maintain sensitive bacterial populations (*Imamovic and Sommer, 2013*; *Kim et al., 2014*), or simultaneously in combination to slow or suppress the evolution of resistance (*Gonzales et al., 2015*; *Munck et al., 2014*; *Rodriguez de Evgrafov et al., 2015*).

Collateral sensitivity to various classes of inhibitors targeting *Plasmodium* parasites has also been observed. For example, isoforms of the chloroquine resistance transporter (*Pf*CRT) that transport chloroquine out of the parasite's digestive vacuole can cause increased sensitivity to a variety of compounds (*Lukens et al., 2014*; *Johnson et al., 2004*; *Sisowath et al., 2009*; *Richards et al., 2016*; *Cooper et al., 2002*; *Evans and Havlik, 1993*; *Small-Saunders et al., 2022*). Collateral sensitivity has also been observed between compounds targeting different subunits of the *Plasmodium* proteasome *Pf*20S (*Kirkman et al., 2018*; *Stokes et al., 2019*), between different chemotypes targeting the P-Type Cation-Transporter ATPase 4 (*Pf*ATP4) (*Flannery et al., 2015*), and between compounds targeting the $Q_o$ and $Q_i$ sites of cytochrome b (*Lukens et al., 2015*). We have also previously shown that collateral sensitivity occurs during the evolution of resistance to *Plasmodium* DHODH inhibitors. Mutations in *dhodh* conferring resistance to one inhibitor can alter the enzyme such that it becomes more sensitive to inhibition by other, structurally distinct compounds (*Lukens et al., 2014*; *Ross et al., 2014*). The existence of collateral sensitivity among a wide range of antimalarial inhibitors suggests a promising opportunity to exploit this phenomenon to block the emergence of resistant parasites (*Lukens et al., 2014*).

In an effort to expand this work, we screened antimalarial libraries from GSK, and successfully identified several compounds with increased activities against DHODH mutants relative to wildtype parasites (*Ross et al., 2018*). In this study, we test the hypothesis that combining two DHODH inhibitors based on their collateral sensitivity profiles would suppress the emergence of resistant parasites. We focused on the compound TCMDC-125334, a DHODH inhibitor that exhibited increased activity against all mutant lines tested, including those that arose during selection with the clinical candidate DSM265. We used in vitro resistance selections to characterize the pathways to resistance for TCMDC-125334 and then tested whether parasites develop resistance in the context of combined treatment with DSM265 and TCMDC-125334.

## Results

### TCMDC-125334 demonstrates activity against all DHODH mutant parasite lines tested

Previously, we screened GSK chemical libraries from the Tres Cantos Antimalarial Set for activity against either wildtype or DHODH mutant parasites (*Supplementary file 1a*; *Ross et al., 2018*). In an extended cross-resistance analysis, we characterized the activity of 17 compounds against three additional DHODH mutant lines, DHODH C276Y, DHODH F227L, and DHODH F227L/L531F (*Figure 1A*, *Figure 1—figure supplement 1*, *Figure 1—source data 1*). We visualized the log-transformed fold change in $EC_{50}$ of each mutant relative to its parental line on a heatmap (*Figure 1B*, *Figure 1—figure supplement 1*, *Figure 1—source data 1*). Hierarchical clustering of this data reveals broad patterns of cross-resistance and collateral sensitivity. The DHODH C276Y, F227I, F227I/L527I, F227L, and F227L/L31F mutant lines have similar dose-response phenotypes. The DHODH E182D and I263F mutant lines also share similar cross-resistance profiles, as we previously reported (*Ross et al., 2018*). In contrast, the DHODH L531F mutant line has a unique sensitivity phenotype (*Figure 1B*). This analysis also suggests some structure-activity relationships. Five related triazolopyrimidine-based compounds (DSM265, DSM267, TCMDC-124402, TCMDC-124417, TCMDC-123826) cluster together, as all mutant lines show cross-resistance against these molecules (*Figure 1B*, *Figure 1—figure supplement 1*, *Figure 1—source data 1*). TCMDC-123823, TCMDC-123545, and TCMDC-123566 also cluster together based on their activity profile, and have a similar chemical structure (*Figure 1B*, *Figure 1—source data 1*). We had previously reported that the compound TCMDC-125334 was active against the DHODH E182E, I263F, F227I, and F227I/L527I lines (*Ross et al., 2018*). In our extended cross-resistance analysis, we showed that TCMDC-125334 is also active against the DHODH C276Y, F227L, and F227L/L531F lines. The eight mutant lines had varying degrees of sensitivity to this compound, from DHODH I263F which is 1.3-fold more sensitive than the wildtype parent, to DHODH C276Y, which is 12-fold sensitive. In contrast, all mutant lines were at least 10-fold resistant to DSM265 (*Figure 1C*).

While none of the mutations in *dhodh* conferred resistance to TCMDC-125334, we previously reported that this compound appears to exclusively target the DHODH enzyme. Expression of the *Saccharomyces cerevisiae* cytosolic type I DHODH (*Sc*DHODH) bypasses the mitochondrial DHODH enzyme, rendering blood-stage parasites insensitive to inhibitors that target electron transport chain proteins DHODH and cytochrome bc1. However, addition of proguanil has a potentiating effect for compounds that target cytochrome bc1, but not DHODH (*Painter et al., 2007*). We previously confirmed that expression of *Sc*DHODH ablates the activity of TCMDC-125334, and that treating the *Sc*DHODH line with TCMDC-125334 and proguanil in combination does not impact this activity (*Ross et al., 2018*; *Dong et al., 2011*).

### Resistance to TCMDC-125334 can be conferred by copy number variation as well as the novel point mutation DHODH I263S

Since TCMDC-125334 showed activity against all DHODH mutant lines tested, we wanted to characterize the evolution of resistance to this compound to determine its independent ability to select resistant mutants. We performed in vitro selections with three independent populations of $10^8$ *Pf*3D7 A10 parasites (*Cowell et al., 2018*). Cultures were treated with 830 nM TCMDC-125334 (approximately the $EC_{99}$) until no living parasites were visible by thin smear microscopy, then allowed to recover in compound-free media. After two rounds of treatment, populations displayed moderate (~2-fold) resistance to TCMDC-125334 (*Figure 2A and B*). Selected parasites were cloned by limiting dilution. Clonal lines isolated from flask 1 had two- or threefold copy number variation (CNV) encompassing the *dhodh* locus, which corresponded to a two- or three-fold resistance phenotype to TCMDC-125334, as well as to the structurally distinct DHODH inhibitors DSM265, Genz669178, and IDI6273. Of the clones isolated from flask 2, four did not have a resistance phenotype, and had no genetic changes in the *dhodh* locus (*Figure 2C–F*, *Table 1*, *Supplementary file 1b*, *Figure 2—figure supplement 1*, *Table 1—source data 1*, *Table 1—source data 2*, *Table 1—source data 3*). One clone, T-F2-C1, was 3-fold resistant to TCMDC-125334 and Genz669178, and 13-fold resistant to IDI-6273 (*Figure 2C–F*, *Table 1*, *Table 1—source data 1*). Whole-genome sequencing (WGS) revealed that clone T-F2-C1 had a point mutation resulting in a DHODH I263S amino acid change (*Table 1—source data 3*). Notably, this DHODH I263S parasite line remained sensitive to DSM265 (*Figure 2D*, *Table 1*, *Table 1—source*

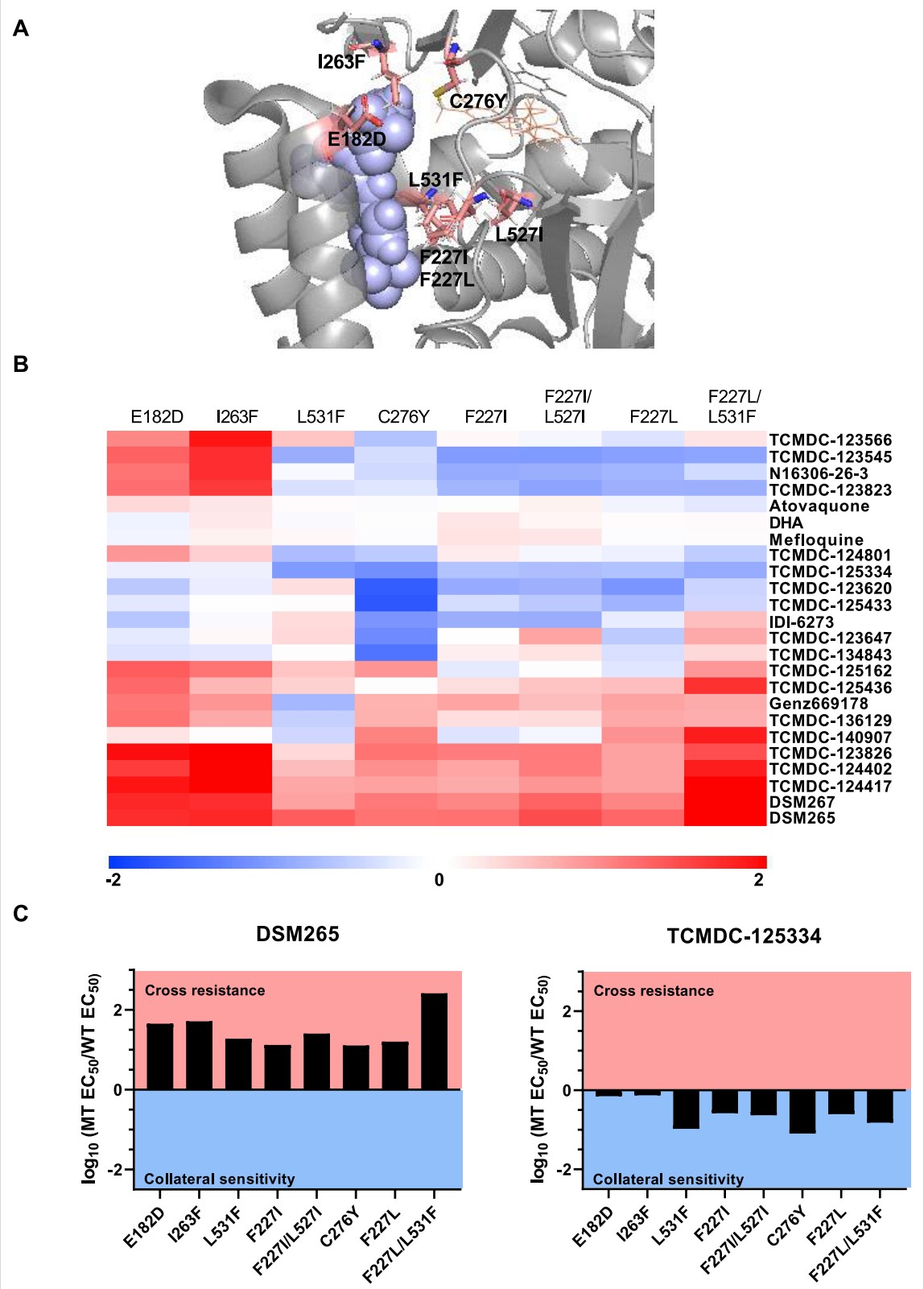

**Figure 1.** Identifying DHODH inhibitors with broad activity against mutant lines. We tested several DSM265-resistant parasite lines against a set of 17 compounds identified from a previous screen of GSK libraries (*Ross et al., 2018*). (**A**) Crystal structure of DHODH bound to DSM265 (PDB ID: 4RX0). Mutations included in our cross-resistance analysis are highlighted in pink. (**B**) To visualize patterns of cross-resistance and collateral sensitivity, we calculated the fold change of each mutant line over its wildtype parent, and plotted the $\log_{10}$-transformed values in a heatmap. Hierarchical clustering

*Figure 1 continued on next page*

Figure 1 continued

by Euclidean distance based on both parasite line and compound was performed using MultiExperimentViewer v4.9. Shades of red indicate that a parasite line is resistant to the indicated compound while blue indicates that it is sensitive. The DHODH C276Y, F227L, and F227L/L531F are newly reported, while other mutant lines were reported in our previous cross-resistance analysis (*Ross et al., 2018*). The wildtype parent for the DHODH F227I, L531F, F227I/L527I, and I263F lines is Dd2. The wildtype parent for the DHODH E182D line is 3D7. The wildtype parent for the DHODH C276Y, F227L, and F227L/L531F lines is 3D7 A10. (**C**) One compound, TCMDC-125334 stands out in this analysis as being active against all mutant lines tested. Shown is a bar graph of the $\log_{10}$ fold change in $EC_{50}$ relative to wildtype for all eight mutant lines tested against DSM265 and TCMDC-125334. DSM265 data was previously reported (*Mandt et al., 2019*). Each dose-response assay was performed with triplicate technical replicates, and average $EC_{50}$'s were obtained from 4 to 5 independent biological replicates. See also *Figure 1—source data 1*.

The online version of this article includes the following source data and figure supplement(s) for figure 1:

**Source data 1.** Individual bioreplicates of $EC_{50}$ values (nM) obtained from dose-response assays.

**Figure supplement 1.** Hierarchical clustering tree of mutant lines and compounds.

*data 1*). (Note that *Figure 2* and *Table 1* show two representative clones from each flask. See *Supplementary file 1b* for a list of all isolated parasite clones.)

In a second independent selection experiment, we treated three additional parasite populations with 1 µM TCMDC-125334. Similarly, bulk populations recovered after either one or two rounds of treatment displayed moderate (~2-fold) resistance to multiple DHODH inhibitors, and exhibited CNV at the *dhodh* locus as detected by qPCR (*Figure 2—figure supplement 1*, *Figure 2—figure supplement 2*). Two of the populations treated with 1 µM were subjected to a second round of treatment at 1 µM TCMDC-125334 for 13 days, then 1.5 µM TMCDC-125334 for 15 days. These populations developed a stronger resistance phenotype (*Figure 2—figure supplement 2*), and correspondingly showed eight-fold increased copy number at the *dhodh* locus (*Figure 2—figure supplement 1*), while no mutations were detected in the *dhodh* locus by WGS (*Figure 2—figure supplement 2—source data 1*).

As a control, we performed a parallel set of selections with 30 nM DSM265 (the $EC_{99}$). Parasites were exposed to compound for 9 days. As previously observed, parasites resistant to DSM265 emerged rapidly after just one round of treatment, after 7 days in compound-free media (*Mandt et al., 2019*). This was expected given that the minimum inoculation of resistance for DSM265 is $10^{5.5}$ (*Duffey et al., 2021*), and we use $10^8$ parasites in our selection experiments. WGS of the bulk populations revealed two point mutations in *dhodh*—DHODH C276Y and I273M (*Figure 2—figure supplement 3*, *Supplementary file 1c*, *Figure 2—figure supplement 3—source data 1*).

## Sequential treatment of DSM265-resistant parasites with TCMDC125334 selects for additional mutations in *dhodh*

Given the hypersensitivity of DHODH C276Y parasites to TCMDC-125334, we also wanted to test what would happen if we treated DHODH C276Y parasites with this compound. In our previous work, selections with two distinct DHODH inhibitors, Genz666136 and DSM74, yielded resistant parasites with a DHODH E182D mutation, which were hypersensitive to the DHODH inhibitor IDI-6273. When we treated the DHODH E182D parasite line with IDI-6273, parasites reverted to the wildtype D182 protein sequence, albeit with a different DNA codon. The D182E revertant line had a sensitivity phenotype similar to wildtype parasites (*Lukens et al., 2014*). We hypothesized that treating DHODH C276Y parasites with TCMDC-125334 would similarly cause a reversion to wildtype, and thus offer a strategy to regain sensitivity to DSM265 (*Figure 3A*).

For these experiments, we used the DHODH C276Y clone that we had previously isolated from in vitro selections with DSM265 (*Mandt et al., 2019*). We treated three 25 mL flasks of approximately $10^8$ DHODH C276Y parasites with 75 nM TCMDC-125334. After parasites recovered, populations were treated with a second round of 100 nM TCMDC-125334. Resistance was observed in all three bulk parasite populations recovered from this second pulse (*Figure 3B and C*). While these parasites were less sensitive to TCMDC-125334 relative to the DHODH C276Y parental line, they were still more sensitive than *Pf*3D7 A10 parasites with a wildtype *dhodh* sequence (*Figure 3C*). Clones isolated from these sequentially selected populations had a similar intermediate dose-response phenotype for TCMDC-125334 (*Figure 3D and E*, *Table 2*, *Supplementary file 1d*, *Table 2—source data 1*). However, compared to the DHODH C276Y parent, they had greatly increased resistance to DSM265, with $EC_{50}$'s>200 nM (*Figure 3E*, *Table 2*, *Supplementary file 1d*, *Table 2—source data 1*). WGS

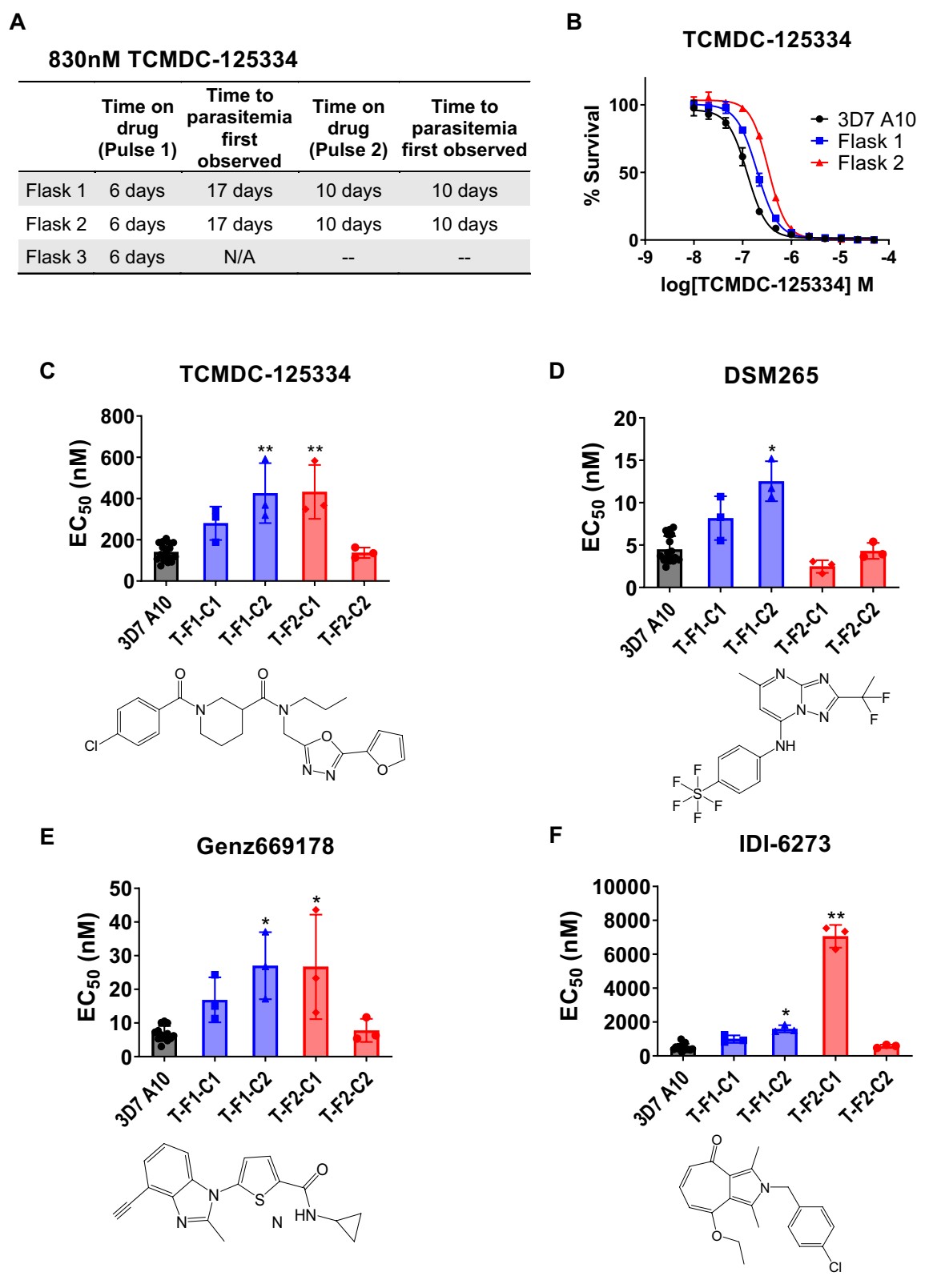

**Figure 2.** In vitro resistance to TCMDC125334 can be mediated by copy number variation as well as the novel point mutation DHODH I263S. (**A**) Protocol for in vitro selection with TCMDC-125334. Parasite populations in three independent 25 mL culture flasks were exposed to 830 nM TCMDC-125334, then allowed to recover in the absence of compound. Resistant parasites emerged after two rounds of treatment within the indicated timeframe. (**B**) In vitro dose-response curve of bulk populations recovered after the second pulse of compound treatment. Error bars show standard

*Figure 2 continued on next page*

*Figure 2 continued*

deviation of three technical replicates. (C–E) Resistant populations were cloned by limiting dilutions. Shown are the average $EC_{50}$ and standard deviation for three biological replicates of dose-response phenotype of two representative clones from each flask for TCMDC-125334 (C), DSM265 (D), Genz669178 (E), and IDI-6273 (F). Each dose-response assay was performed with triplicate technical replicates, and average $EC_{50}$'s were obtained from 3 to 4 independent biological replicates. Compound structures for TCMDC-125334, DSM265, Genz669178, and IDI-6273 are displayed below their corresponding graphs. The parasite lines are each designated with a unique identifier; for example, T-F1-C1 was selected with TCDC-125334 (T) came from flask 1 (F1), and is designated as 'C1' for 'Clone 1'. DHODH genotypes for representative clones are as follows: T-F1-C1: 2x *dhodh* copy number variation (CNV); T-F1-C2: 3x *dhodh* CNV; T-F2-C1: DHODH I263F; T-F2-C1: WT. Statistical significance was determined by a Kruskal-Wallis test, with post hoc multiple comparisons (Dunn's) of each clone to 3D7 A10. *p≤0.05; **p≤0.01. See also *Table 1*, *Supplementary file 1b*, and *Table 1—source data 1*.

The online version of this article includes the following source data and figure supplement(s) for figure 2:

**Figure supplement 1.** Copy number variation of the *dhodh* locus in 3D7 A10 parasites selected with TCMDC-125334.

**Figure supplement 2.** Treatment with 1 µM TCMDC-125334 selects for moderately resistant parasite populations with copy number variation at the *dhodh* locus.

**Figure supplement 2—source data 1.** Whole-genome sequencing analysis of bulk selected populations.

**Figure supplement 3.** Control selection with DSM265.

**Figure supplement 3—source data 1.** Whole-genome sequencing analysis of bulk selected populations.

revealed that parasites retained the DHODH C276Y mutation, and gained additional mutations in *dhodh*, rather than reverting back to wildtype (*Table 2*, *Table 2—source data 2*, *Table 2—source data 3*). Clones isolated from flask 1 and flask 2 had increased *dhodh* copy number, in addition to the DHODH C276Y mutation, as confirmed by qPCR (*Table 2*, *Figure 3—figure supplement 1*, *Table 2—source data 2*). Clones from flask 3 had gained an additional F227Y mutation (*Table 2*, *Table 2—source data 3*). (Note that *Figure 3* and *Table 2* show two representative clones from each flask. See *Supplementary file 1d* for a list of all isolated parasite clones.) Overall, sequential selection of a DSM265-resistant line with TCMDC-125334 does not restore sensitivity to DSM265. While sequentially selected parasites with additional mutations in *dhodh* were highly resistant to DSM265, they were not cross-resistant to TCMDC-125334, suggesting that they would still be sensitive to a therapeutic dose.

**Table 1.** DHODH genotype and corresponding dose-response phenotype for representative in vitro TCMDC-125334 selected lines.

| Clone ID | DHODH mutation(s)* | DHODH CNV | TCMDC-125334 EC50 (nM) | DSM265 EC50 (nM) | IDI-6273 EC50 (nM) | Genz669178 EC50 (nM) |
|---|---|---|---|---|---|---|
| 3D7 A10 | WT | 1 | 140±42.6 | 4.5±1.6 | 502±209 | 6.9±2.2 |
| T-F1-C1 | WT | 2 | 280±81.3 | 8.2±2.6 | 993±220 | 16.9±6.70 |
| T-F1-C2 | WT | 3 | 426±146** | 12.5±2.36* | 1597±212* | 27.0±10.0* |
| T-F2-C1 | I263S | 1 | 432±130.8** | 2.5±0.74 | 7057±671** | 26.7±15.5* |
| T-F2-C2 | WT | 1 | 137±25.1 | 4.3±0.94 | 572±91.5 | 7.79±3.42 |
| Overall p-value (approximate) | | | 0.0004 | 0.0013 | 0.0007 | 0.0013 |
| Kruskall-Wallis statistic | | | 28.52 | 25.54 | 27.05 | 25.50 |

The parasite lines are each designated with a unique identifier; for example, T-F1-C1 was selected with TCDC-125334 (T) came from flask 1 (F1), and is designated as 'C1' for 'Clone 1'. Data is shown as mean $EC_{50}$ ± standard deviation. Statistical significance was determined by a Kruskal-Wallis test, with post hoc multiple comparisons (Dunn's) of each clone to 3D7 A10. *p≤0.05; **p≤0.01. Overall statistics are reported for each comparison group. Each dose-response assay was performed with triplicate technical replicates, and average $EC_{50}$'s were obtained from 3 to 4 independent biological replicates.
*Variants identified by whole-genome sequencing.

The online version of this article includes the following source data for table 1:

**Source data 1.** Individual bioreplicates of $EC_{50}$ values (nM) obtained from dose-response assays.

**Source data 2.** Copy number variation (CNV) analysis based on whole-genome sequencing.

**Source data 3.** Homozygous variants identified from in vitro selections by whole-genome sequencing.

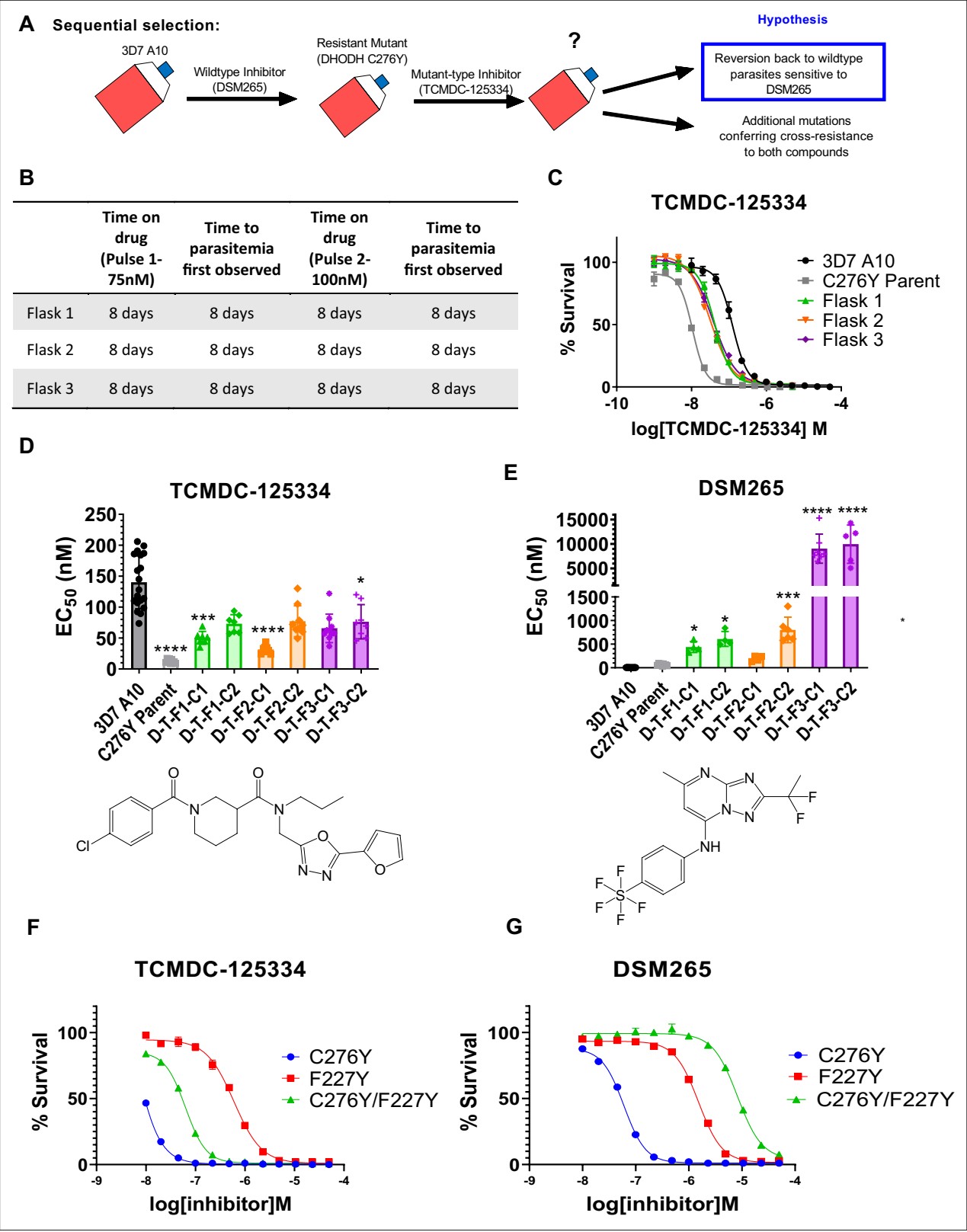

**Figure 3.** Mutant parasites can overcome collateral sensitivity by acquiring additional genetic changes that confer high-level resistance to wildtype inhibitors. (**A**) General schematic for sequential selection. (**B**) Protocol for in vitro selection. DHODH C276Y mutant parasites in three independent 25 mL culture flasks were exposed to 40 nM TCMDC-125334, then allowed to recover. The time on compound and time to recovery are indicated. After two rounds of treatment and recovery, resistant parasites are observed in all three flasks. (**C**) In vitro dose-response curve of selected bulk populations.

*Figure 3 continued on next page*

*Figure 3 continued*

Error bars show standard deviation of technical replicates. (**D–E**) Resistant populations were cloned by limiting dilution. Shown is the average $EC_{50}$ of two representative clones from each flask for TCMDC-125334 (**C**), and DSM265 (**D**), with chemical structures illustrated below. Bar graphs represent average $EC_{50}$ with error bars depicting standard deviation. Individual biological replicates are also shown. Each dose-response assay was performed with triplicate technical replicates, and average $EC_{50}$'s were obtained from 3 to 8 independent biological replicates. The parasite lines are each designated with a unique identifier; for example, D-T-F1-C1 was selected first with DSM265 (**D**), then from TCMDC-125334 (**T**), came from flask 1 (**F1**), and is designated as 'C1' for 'Clone 1'. DHODH genotypes for representative clones are as follows: D-T-F1-C1: DHODH C276Y+3x *dhodh* copy number variation (CNV); D-T-F1-C2: DHODH C276Y+4x *dhodh* CNV; D-T-F2-C1: DHODH C276Y+2x *dhodh* CNV; D-T-F2-C2: DHODH C276Y+4x *dhodh* CNV; D-T-F3-C1: DHODH C276Y/F227Y; D-T-F3-C2: DHODH C276Y/F227Y. Statistical significance was determined by a Kruskal-Wallis test, with post hoc multiple comparisons (Dunn's) of each clone to 3D7 A10. In cases where the EC50 could not be determined with range of concentrations tested, the maximum concentration that achieved >50% growth was used as a stand-in value. *p≤0.05; ***p≤0.001; ****p≤0.0001. See also *Table 2*, *Supplementary file 1d*, and *Table 2—source data 1*. (**F–I**) The DHODH C276Y/F227Y double mutant exhibits a resistant phenotype intermediate to single mutations. The dose-response phenotypes of the DHODH C276Y/F227Y line selected in this study (clone C276Y T-F3-C3), the DHODH C276Y line previously selected with DSM265, and the DHODH F227Y line previously selected with DSM267 (*Mandt et al., 2019*) were characterized. Shown is a representative dose-response curve for DSM265 (**F**) and TCMDC-125334 (**G**) for illustration. Error bars show standard deviation of technical replicates within a single assay.

The online version of this article includes the following figure supplement(s) for figure 3:

**Figure supplement 1.** Copy number variation of the *dhodh* locus in DHODH C276Y parasites selected with TCMDC-125334.

We wanted to further characterize how the DHODH C276Y and DHODH F227Y mutations were interacting to contribute to the observed dose-response phenotypes. Compared to wildtype, the DHODH C276Y mutation alone is ~10-fold resistant to DMS265, and ~10-fold sensitive to TCMDC-125334 (*Table 3*). In contrast, the DHODH C276Y/F227Y double mutant is ~1000-fold resistant to

**Table 2.** DHODH genotype and corresponding dose-response phenotype for representative in vitro selection of DHODH C276Y parent with TCMDC-125334.

| Clone ID | DHODH mutation(s)* | DHODH CNV | TCMDC-125334 EC50 (nM) | DSM265 EC50 (nM) | IDI-6273 EC50 (nM) | Genz669178 EC50 (nM) |
|---|---|---|---|---|---|---|
| 3D7 A10 | WT | 1 | 140±42.6 | 4.5±1.6 | 502±209 | 6.9±2.2 |
| C276Y Parent | C276Y | 1 | 12.40±3.089**** | 63.9±19.84 | 41.7±12.7**** | 22.0±6.68 |
| D-T-F1-C1 | C276Y | 3–4 | 50.3±10.4*** | 439.3±115.96* | 185±36.1 | >106 |
| D-T-F1-C2 | C276Y | 4 | 73.0±14.7 | 611±159* | 345±64.1 | >106 |
| D-T-F2-C1 | C276Y | 2 | 31.1±6.89**** | 196±56.6 | 138±44.0* | 117±30.0 |
| D-T-F2-C2 | C276Y | 4 | 77.2±25.1 | 801±270 | 326±155 | ND |
| D-T-F3-C1 | C276Y/F227Y | 1 | 65.6±22.9* | 9042±2985**** | 163±36.3** | 69.7±14.5 |
| D-T-F3-C2 | C276Y/F227Y | 1 | 76.3±27.7 | 9960±3923**** | 183±84.3 | 97±54.5 |
| Overall p-value (approximate) | | | <0.0001 | <0.0001 | <0.0001 | <0.0001 |
| Kruskall-Wallis statistic | | | 38.12 | 47.36 | 45.55 | 54.32 |

The parasite lines are each designated with a unique identifier; for example, D-T-F1-C1 was selected first with DSM265 (D), then from TCMDC-125334 (T), came from flask 1 (F1), and is designated as 'C1' for 'Clone 1'. Data is shown as mean $EC_{50}$ ± standard deviation. Statistical significance was determined by a Kruskal-Wallis test, with post hoc multiple comparisons (Dunn's) of each clone to 3D7 A10. *p≤0.05; **p≤0.01; ***p≤0.001; ****p≤0.0001. Overall statistics are reported for each comparison group. Each dose-response assay was performed with triplicate technical replicates, and average $EC_{50}$'s were obtained from 3 to 8 independent biological replicates. WT = wildtype. ND indicates that the $EC_{50}$ could not be determined; parasites were resistant to the range of doses tested as indicated by lack of complete kill at the highest dose of 500 nM. A representative set of clones were tested at higher concentrations.
*Variants identified by whole-genome sequencing.

The online version of this article includes the following source data for table 2:

**Source data 1.** Individual bioreplicates of EC50 values (nM) obtained from dose-response assays.

**Source data 2.** Copy number variation (CNV) analysis based on whole-genome sequencing.

**Source data 3.** Homozygous variants identified from in vitro selections by whole-genome sequencing.

**Table 3.** Resistance phenotype of C276Y and F227Y single mutants and C276Y/F227Y double mutant compared to expected phenotype under additive epistasis.

| DHODH mutations | TCMDC-125334 EC50 (fold change) | DSM265 EC50 (fold change) | IDI-6273 EC50 (fold change) | Genz669178 EC50 (fold change) |
|---|---|---|---|---|
| 3D7 A10 | 140±42.6 | 4.5±1.6 | 502±209 | 6.9±2.2 |
| C276Y | 12.40±3.09 (0.089) | 63.9±19.8 (14.2) | 41.7±12.7 (0.083) | 22.0±6.68 (3.2) |
| F227Y | 408±17.6 (2.9) | 998±50.0 (221) | 2210±289 (4.4) | 18.8±3.39 (2.7) |
| C276Y/F227Y (D-T-F3-C1) | 65.6±22.9 (0.46) | 9042±2985 (2009) | 163±36.3 (0.32) | 69.7±14.5 (10.1) |
| Expected fold change for C276Y/F227Y (additive assumption) | 0.26 | 3138 | 0.37 | 8.64 |

$EC_{50}$ (nM) of each line is reported, with fold change relative to 3D7 A10 in parentheses. The expected fold change of the C276Y double mutant based on an assumption of additive epistasis is reported for comparison.

The online version of this article includes the following source data for table 3:

**Source data 1.** Individual bioreplicates of EC50 values (nM) obtained from dose-response assays.

DSM265. However, the double mutant line is still ~2-fold sensitive to TCMDC-125334 relative to wildtype. We had previously selected the DHODH F227Y single mutation with the triazolopyrimidine-based inhibitor DSM267, but we had not yet assessed this mutant line's sensitivity to TCMDC-125334. We characterized the dose-response phenotype of the DHODH F227Y mutant line and found that it exhibited three-fold resistance to TCMDC-125334 (*Table 3*, *Table 3—source data 1*). Comparing the dose-response phenotype of the single mutant lines DHODH C276Y and DHODH F227Y to the phenotype of the DHODH C276Y/F227Y double mutant, we find that the phenotype of the two mutations in combination is consistent with additivity, ruling out a negative epistatic interaction (*Figure 3F–I*, *Table 3*).

## Treatment with TCMDC-125334 + DSM265 combination delays but does not prevent the emergence of resistance

Based on the collateral sensitivity observed between DSM265 and TCMDC-125334, we hypothesized that treatment with a combination of DSM265 and TCMDC-125334 would suppress the emergence of resistant parasites (*Figure 4A*). We treated three independent populations of $10^8$ *Pf*3D7 A10 parasites with both compounds simultaneously, using the $EC_{99}$ of each. The selected parasite populations were not resistant after one or two rounds of treatment. However, after three pulses of compound treatment totaling 63–73 days, we obtained two populations cross-resistant to both compounds (*Figure 4B and C*). Three of the four clonal lines isolated from flask 1 were moderately (~2-fold) resistant to both DSM265 and TCMDC-125334, as well as Genz669178 and IDI-6273 (*Figure 4D–G*, *Table 4*, *Supplementary file 1e*, *Table 4—source data 1*). Copy number duplication of the *dhodh* locus was detected in these resistant clones by qPCR and WGS (*Table 4*, *Figure 4—figure supplement 1*, *Table 4—source data 2*). All six clones isolated from flask 2 were four- to sixfold resistant to TCMDC-125334 and 16- to 20-fold resistant to DSM265. These clones were also ~2-fold resistant to IDI-6273, but were still sensitive to Genz669178 (*Figure 4D–G*, *Table 4*, *Supplementary file 1e*, *Table 4—source data 1*). Sequencing revealed that these six clones had the same point mutation resulting in a DHODH V532A amino acid change (*Table 4—source data 3*, *Table 4—source data 4*). (Note that *Figure 4* and *Table 4* show two representative clones from each flask. See *Supplementary file 1e* for a list of all isolated parasite clones.) Thus, although resistance took longer to emerge with combination treatment compared to treatment with DSM265 or TCMC-125334 alone (63–73 days with combination treatment vs. 43 days with TCMDC-125334 alone and 16 days with DSM265 alone), this strategy was ultimately insufficient to prevent the emergence of parasites resistant to both compounds. This result would argue against using these compounds in combination.

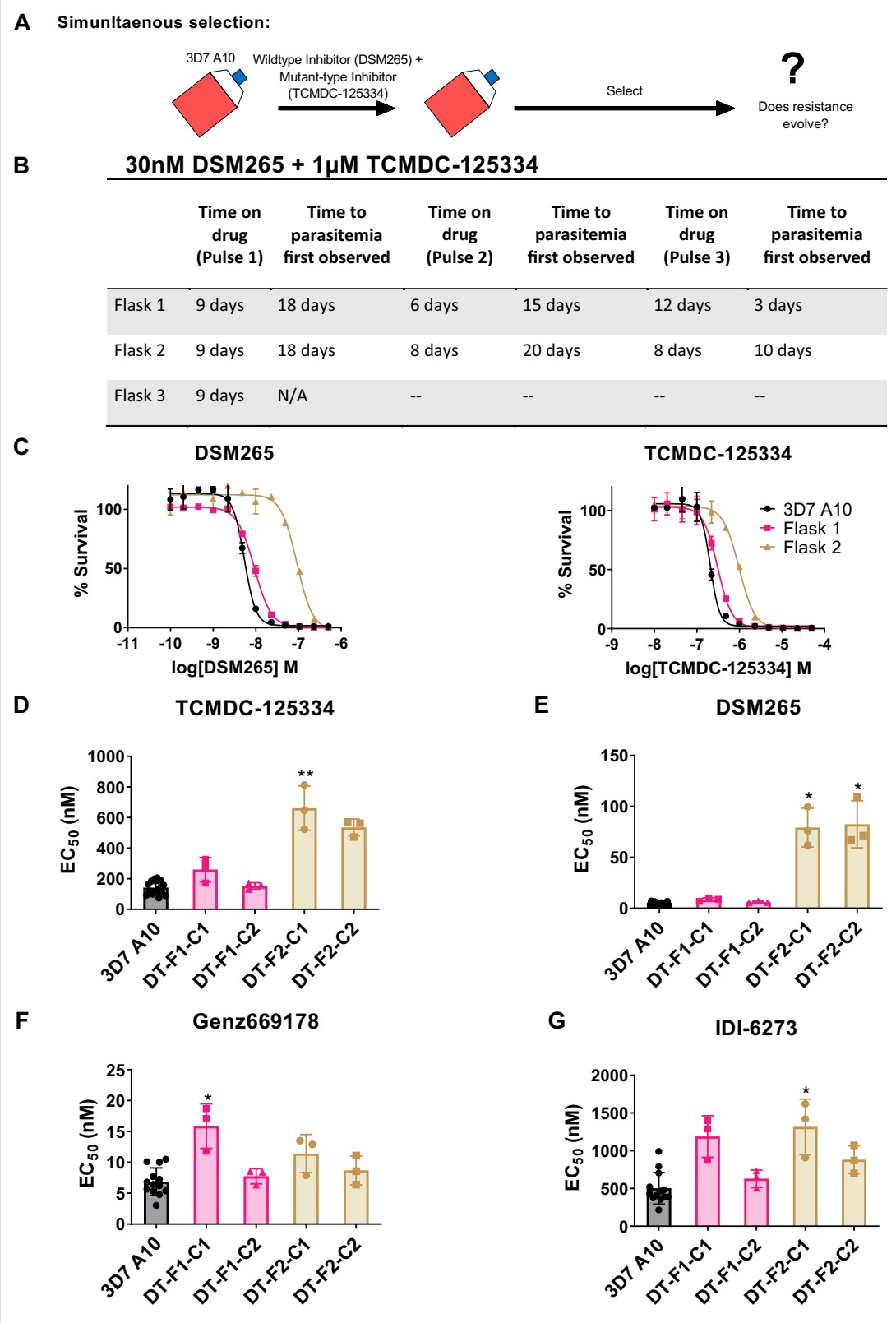

**Figure 4.** Resistance to the DSM265 + TCMDC-125334 combination arose after three rounds of treatment, with cross-resistance to both compounds conferred by the novel DHODH V532A mutation. (**A**) General schematic for simultaneous selection. (**B**) Protocol for in vitro selection with TCMDC-125334 and DSM265. Parasite populations in three independent 25 mL culture flasks were exposed to combination treatment, then allowed to recover. The time on compound and time to recovery are indicated. After three rounds of treatment and recovery, resistant parasites are observed

*Figure 4 continued on next page*

*Figure 4 continued*

in flasks 1 and 2. (**C**) In vitro dose-response curve of bulk populations from flasks 1 and 2. Error bars show standard deviation of technical replicates. (**D–G**) Resistant populations were cloned by limiting dilutions. Shown is the average $EC_{50}$ of two representative clones from each flask for TCMDC-125334 (**D**), DSM265 (E), Genz669178 (F), and IDI6273 (G). Bar graphs represent average $EC_{50}$ with error bars depicting standard deviation. Individual biological replicates are also shown. Each dose-response assay was performed with triplicate technical replicates, and average $EC_{50}$'s were obtained from 3 independent biological replicates. The parasite lines are each designated with a unique identifier; for example, DT-F1-C1 was selected with DSM255 and TCDC-125334 (DT) came from flask 1 (**F1**), and is designated as 'C1' for 'Clone 1'. Statistical significance was determined by a Kruskal-Wallis test, with post hoc multiple comparisons (Dunn's) of each clone to 3D7 A10. *p≤0.05; **p≤0.01. See also *Table 3*, *Supplementary file 1e*, and *Table 3—source data 1*.

The online version of this article includes the following figure supplement(s) for figure 4:

**Figure supplement 1.** Copy number variation of the *dhodh* locus in 3D7 A10 parasites selected with DSM265 + TCMDC-125334 simultaneously.

## In silico docking of TCMDC-125334 to *Pf*DHODH suggests molecular mechanisms of resistance

To better understand the binding mode of TCMDC-125334 and develop inferences on the molecular mechanisms by which mutations characterized in this study may confer resistance or hypersensitivity to the compound, we performed in silico molecular docking using Flare, v.4.0.3.40719 (Cresset-Group). The co-crystal structures of DHODH, flavin mononucleotide (FMN), orotate, and a variety of inhibitors have previously been reported, but there is no reported structure with TCMDC-125334 (*Phillips et al., 2015*; *White et al., 2019*; *Ross et al., 2014*; *Booker et al., 2010*). The best pose predicted by our simulations shows that TCMDC-125334 docks within same hydrophobic binding pocket as DSM265 and other DHODH inhibitors (*Figure 5A and B*). As with all published crystal structures of DHODH inhibitors, our simulations suggest that TCMDC-125334 binds allosterically with regard to both orotate/dihydroorotate and FMN.

Our docking results suggest that the DHODH[V532A] mutation would disrupt hydrophobic interactions with the furan and 1,3,4-oxadiazole rings, consistent with the observed 5.3-fold resistance seen with the DHODH[V532A] mutants. There are no published crystal structures for the DHODH[C276Y] so we used

**Table 4.** DHODH genotype and corresponding dose-response phenotype for representative in vitro TCMDC-125334 + DSM265 selected lines.

| Clone ID | DHODH mutation(s) | DHODH CNV | TCMDC-125334 EC50 (nM) | DSM265 EC50 (nM) | IDI-6273 EC50 (nM) | Genz669178 EC50 (nM) |
|---|---|---|---|---|---|---|
| 3D7 A10 | WT | 1 | 140±42.6 | 4.5±1.6 | 502±209 | 6.9±2.2 |
| DT-F1-C1 | WT* | 2 | 260±79.5 | 8.6±1.6 | 1190±276 | 15.9±3.61* |
| DT-F1-C2 | WT[†] | 1 | 153±21.0 | 6.1±0.88 | 628±117 | 7.74±1.23 |
| DT-F2-C1 | V532A* | 1 | 661±145** | 79.2±19.0* | 1320±368* | 11.4±3.08 |
| DT-F2-C2 | V532A* | 1 | 536±54.4 | 82.4±23.1* | 881±185 | 8.69±2.35 |
| Overall p-value (approximate) | | | <0.0001 | <0.0001 | 0.0019 | 0.0091 |
| Kruskall-Wallis statistic | | | 35.72 | 35.80 | 27.85 | 23.50 |

The parasite lines are each designated with a unique identifier; for example, DT-F1-C1 was selected with DSM255 and TCDC-125334 (DT) came from flask 1 (F1), and is designated as 'C1' for 'Clone 1'. Data is shown as mean $EC_{50}$ ± standard deviation. Statistical significance was determined by a Kruskal-Wallis test, with post hoc multiple comparisons (Dunn's) of each clone to 3D7 A10. *p≤0.05; **p≤0.01. Overall statistics are reported for each comparison group. Each dose-response assay was performed with triplicate technical replicates, and average $EC_{50}$'s were obtained from 3 independent biological replicates. WT = wildtype.

*DHODH genotype determined by whole-genome sequencing.

[†]DHODH genotype determined by Sanger sequencing.

The online version of this article includes the following source data for table 4:

**Source data 1.** Individual bioreplicates of EC50 values (nM) obtained from dose-response assays.

**Source data 2.** Copy number variation (CNV) analysis based on whole-genome sequencing.

**Source data 3.** Homozygous variants identified from in vitro selections by whole-genome sequencing.

**Source data 4.** Sanger sequencing of in vitro selected clones selected with DSM265 + TCMDC-125334.

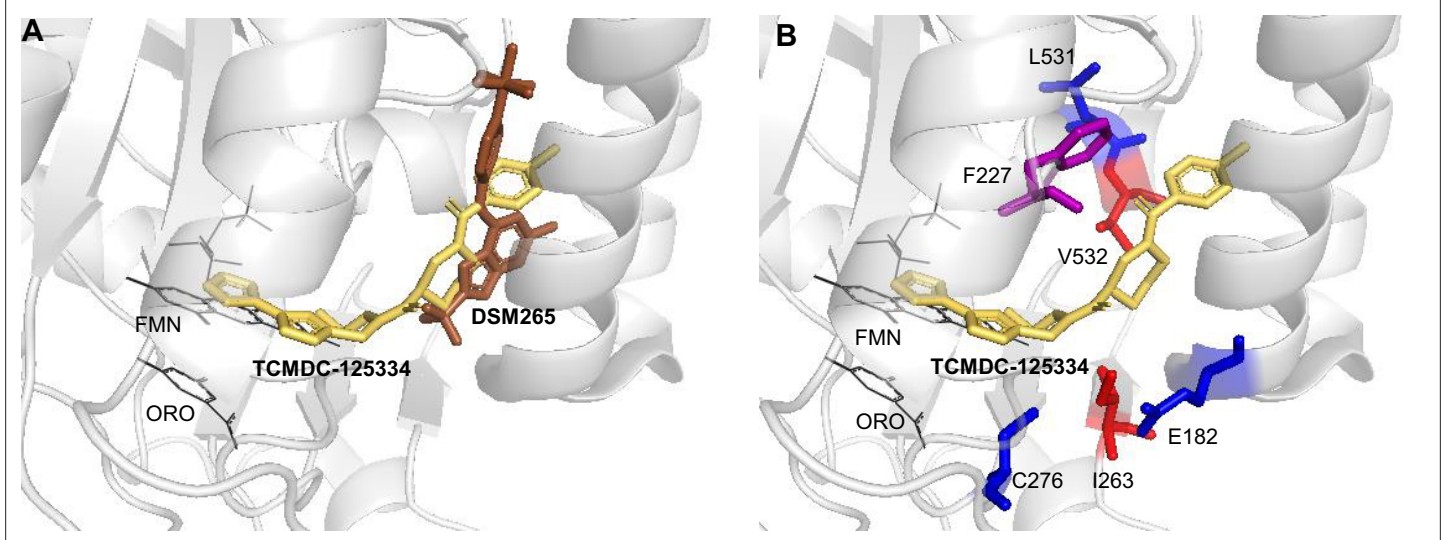

**Figure 5.** Molecular docking of TCMDC-125334 with *Pf*DHODH. (**A**) TCMDC-125334 docks within the inhibitor binding pocket of DHODH. Shown is overlay of TCMDC-125334 docked in combined structure, and DSM265 from the 4XR0 structure. (**B**) Mutations conferring cross-resistance or collateral sensitivity are in close proximity to TCMDC-125334. Key residues discussed in the study are highlighted. Residues conferring cross-resistance are shown in red, while residues conferring collateral sensitivity are in blue. The F227 residue is colored purple, as the F227L and F227I substitutions confer collateral sensitivity while the F227Y substitution confers resistance. Structure labeled ORO is the orotic acid product. FMN is the flavin mononucleotide cofactor.

Flare's mutate function to produce a model of DHODH$^{C276Y}$, optimized the results in the absence of other molecules and then in the presence of TCMDC-125334, orotate, and FMN with Flare's protein minimize function, and rescored the result. Our simulation predicted that the DHODH$^{C276Y}$ mutant would have higher affinity for TCMDC-125334 than wildtype.

### Cross-resistant parasites are relatively fit in in vitro growth assays

Given that the DHODH V532A mutation confers cross-resistance to DSM265 and TCMDC-125334, we wanted to assess the relative fitness of *P. falciparum* parasites carrying this mutation. Previously we have shown that some *dhodh* mutations, such as DHODH E182D, can negatively impact enzyme function and in vitro growth, while other mutations, such as DHODH C276Y, do not appear to confer a fitness cost (*Mandt et al., 2019*; *Ross et al., 2014*). Because resistance to simultaneous selection with DSM265 and TCMDC-125334 took longer to emerge, we hypothesized that DHODH V532A mutants might have a growth defect relative to wildtype parasites. To test this hypothesis, we performed in vitro competitive growth assays. Synchronized DHODH V532A and wildtype 3D7 A10 lines were mixed in a 1:1 ratio. The mixed culture was grown in three independent flasks over a 4-week period in compound-free media, and genomic DNA was collected every 7–9 days. The ratio of mutant and wildtype parasites was determined by calculating the percentage of reads detecting the V532A allele in WGS. This analysis showed that the DHODH V532A variant remained at ~50% frequency over time (*Figure 6*, *Supplementary file 1*). As a confirmation of this result, we also assessed the resistance phenotype of mixed cultures. On day 0, the mixed culture exhibited an intermediate dose-response phenotype, as illustrated by a biphasic dose-response curve. This intermediate phenotype was maintained at day 28 (*Figure 6—figure supplement 1*). Overall, both the genotypic and phenotypic data indicate that the DHODH V532A mutant is as fit as wildtype parasites.

### Discussion

The main conclusion of our study is that combining two DHODH inhibitors, DSM265 and TCDC-125334, based on their collateral sensitivity profiles fails to prevent the emergence of resistance in *P. falciparum* in vitro. Although all previously identified mutations that we tested exhibited sensitivity to TCMDC-125334, we identified the point mutation DHODH I263S in selection with TCMDC-125334 alone, which was resistant to TCMDC-125334 but not DSM265. We found that sequentially treating

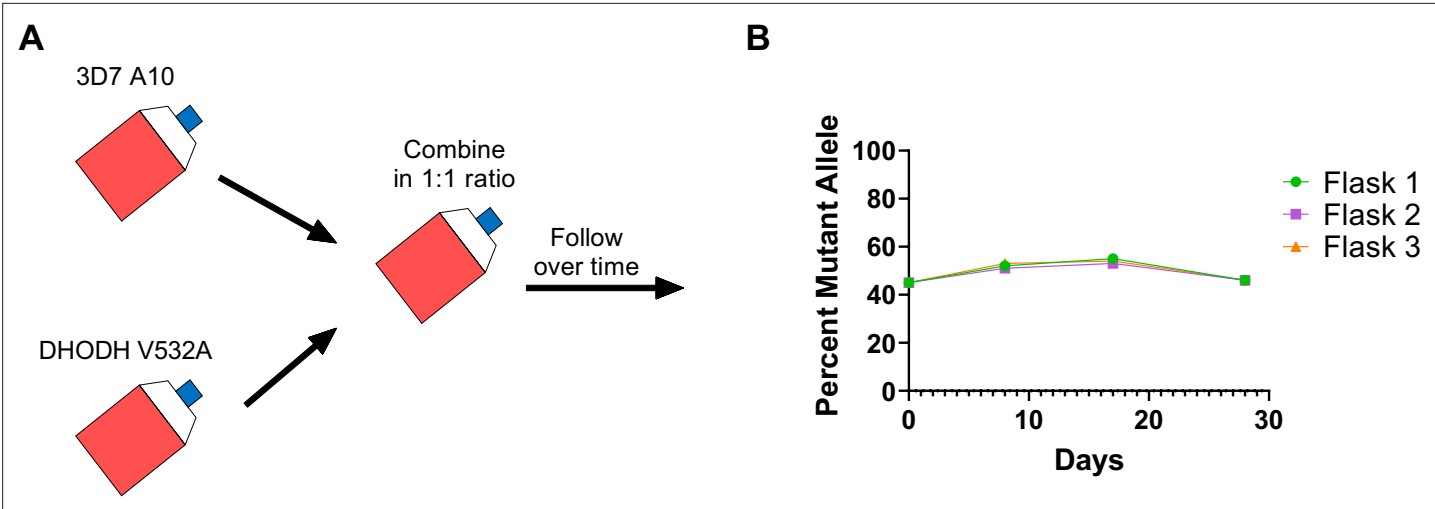

**Figure 6.** The DHODH V532A mutation does not confer a fitness cost in in vitro competitive growth assays. (**A**) Schematic of competitive growth experiments. Synchronized 3D7 parent and DHODH V532A mutant parasites at 1% ring-stage parasitemia were mixed at equal ratios. The mixed culture was then split into three independent flasks, and followed over time. Mixed cultures were grown for 4 weeks, and genomic DNA was collected every 7–9 days. (**B**) The percent mutant allele was calculated at each timepoint based on whole-genome sequencing reads (*Supplementary file 1f*).

The online version of this article includes the following figure supplement(s) for figure 6:

**Figure supplement 1.** Co-cultured parasites in competition assay retain an intermediate phenotype.

a collaterally sensitive mutant line with TCMDC-125334 selected for parasites with additional genetic changes in *dhodh*, including the DHODH F227Y mutation, which alone also confers resistance to TCMDC-125334. While the DHODH C276Y/F227Y double mutant is less sensitive than the DHODH C276Y parent, it is still more sensitive to TCMDC-125334 compared to wildtype parasites. Finally, we found that treatment with DSM265 and TCMDC-125334 in combination selected for cross-resistant parasites with a DHODH V532A mutation. We also demonstrated that the DHODH V532A mutation does not exhibit a fitness cost in in vitro competitive growth assays.

While we identified three point mutations in this study that confer resistance to TCMDC-125334, the majority of *dhodh* mutations tested thus far are sensitive to this molecule. CNV was a common mechanism of resistance to TCMDC-125334 across multiple independent selections. Three of four populations of wildtype 3D7 A10 selected with TCMDC-125334 exhibited increased copy number of *dhodh*. When we selected the collaterally sensitive DHODH C276Y line with TCMDC-125334, clones isolated from two of three independent populations exhibited two- to fourfold CNV of *dhodh*. In contrast to this, the majority of in vitro selections with DSM265 and DSM267 selected for point mutations in the *dhodh* locus, with CNVs occurring only seldomly (*Mandt et al., 2019*). While CNVs confers cross-resistance across various DHODH inhibitor compound classes, this mechanism confers only moderately reduced susceptibility. CNVs are also less evolutionarily stable compared to point mutations. Others have shown that additional copies of *dhodh* are lost in in vitro culture over time (*Guler et al., 2013*).

Our finding that sequentially treating the DHODH C276Y line with a second inhibitor led to the acquisition of additional genetic changes, rather than reversion back to wildtype, has important implications for treatment strategies to manage resistance. This result is in contrast with our previous work, in which selecting DHODH E182D parasites with the mutant-type inhibitor IDI-6273 caused parasites to revert back to the wildtype amino acid sequence. Interestingly, we also observed that the DHODH E182D parasite had a competitive fitness defect (*Lukens et al., 2014*; *Ross et al., 2014*), while we previously showed that the DHODH C276Y is as fit as wildtype in in vitro competitive growth assays (*Mandt et al., 2019*). Previous research on collateral sensitivity in *Pseudomonas aeruginosa* from Barbosa et al. suggests that differing evolutionary outcomes (reversion vs. secondary mutation) may be due to differences in the fitness cost of the initial resistance mutations (*Barbosa et al., 2019*). Similarly, we saw that the competitively fit DHODH C276Y line favored alternative pathways to cross-resistance rather than reversion to wildtype. Our work has implications for strategies such as drug

cycling or the simultaneous use of multiple first-line treatments that aim to delay the emergence of resistance (*Boni et al., 2008*; *Huijben and Paaijmans, 2018*). Such strategies assume that resistance mutations will disappear in the absence of pressure from the original selecting agent. However, our results suggest that treatment with a second compound can also result in the acquisition of additional genetic changes and subsequent multidrug resistance.

Comparing the phenotype of the DHDOH C276Y/F227Y double mutant with DHODH C276Y and DHODH F227Y parasite lines, we also note that the two mutations in combination exhibit additive epistasis. This resulted in markedly high-level resistance to DSM265 (>1000-fold relative to wild-type). Multiple studies indicate that epistasis is an important factor in determining the evolution of resistance and cross-resistance/collateral sensitivity (*Rosenkilde et al., 2019*; *Barbosa et al., 2019*; *Apjok et al., 2019*; *Lozovsky et al., 2009*; *Dellus-Gur et al., 2015*). Negative epistasis between resistance mutations within the same enzyme constrains the number of pathways to higher-level resistance (*Lozovsky et al., 2009*; *Dellus-Gur et al., 2015*). In the context of collateral sensitivity, the study from *Barbosa et al., 2019* also indicates that re-sensitization to wildtype is favored when there is negative epistasis between resistance mechanisms. In contrast, our result highlights how, under additive epistasis, the emergence of an initial resistance mutation can open up pathways to higher-level resistance. The major take-away of this study is that combination treatment with DSM265 and TCMDC-125334 failed to suppress resistance, and we hypothesize that this is due to the great diversity of evolutionarily competitive *dhodh* mutations that confer resistance to various inhibitors. Other reports in bacteria have also found that observed patterns of collateral sensitivity are often thwarted due to the existence of diverse trajectories to resistance. When selections are repeated enough times (*Nichol et al., 2019*), or are performed with a large enough starting populations (*Jiao et al., 2016*), cross-resistant variants eventually emerge, even if they are relatively rare. Similarly, we find that although most of the mutations we originally identified were collaterally sensitive to TCMDC-125334, repeated treatment with DSM265 and TCMDC-125334 in combination ultimately yielded the cross-resistant variant DHODH V532A. There were also additional mutations outside the *dhodh* locus observed in the WGS analysis that were not repeated across multiple independent selections and so are less likely to contribute to the resistance phenotype (*Table 1—source data 3*; *Table 2—source data 3*; *Table 4—source data 3*). Additionally, we have previously conducted allelic replacements with mutations in DHODH which have phenocopied the selected lines, suggesting that these mutations are the primary driver of the observed resistance phenotypes (*Mandt et al., 2019*).

We cannot quantify the frequency of resistance emergence based on our results, which would be necessary to fully understand the risk of resistance to DSM265 + TCMDC-125334 in combination compared to treating with either compound alone. However, we did find that the DHODH V532A mutation conferring cross-resistance to both compounds emerged from a population of $10^8$ parasites after three treatment cycles. In contrast, there can be $10^{10}$–$10^{13}$ parasites in an infected human host, suggesting that this would be a sufficient population size for cross-resistance to emerge (*Goldberg et al., 2012*; *Bopp et al., 2013*).

This study also highlights the flexibility of the DHODH enzyme. Prior to this work, we had identified 13 individual point mutations and 2 sets of double mutations that conferred resistance to DHODH. In this study, we selected two additional point mutations—DHODH I263S and DHODH V532A—as well as the DHODH C276Y/F227Y double mutant line. Interestingly, the DHODH I263S mutation is not resistant to DSM265, while the DHODH I263F mutation at the same site confers strong (~100-fold) resistance to this compound. Previous studies have crystallized the DHODH enzyme bound to various inhibitors, including DMS265 (PDB 4*R*X0), Genz669178 (PDB 4CQ8), and IDI-6273 (PDB 4CQA) (*Phillips et al., 2015*; *Lukens et al., 2014*). These compounds all bind within the same flexible pocket (see *Figure 1A*) of the enzyme. Our results demonstrate that this pocket not only accommodates the binding of a variety of chemical structures, but is also very mutationally flexible, highlighting the resistance liability of this enzyme as a drug target. Molecular simulation studies provided insight into the potential mechanism of resistance to both DSM265 and TCMDC-125334, highlighting the interactions at position V532. However, understanding the mechanism of collateral sensitivity will require further study. Molecular simulation points to differential binding affinity of TCMDC-125334 to the wildtype and C276Y mutant proteins, consistent with the increased sensitivity of the mutant observed in our study (*Figure 5*).

The in vitro studies described here also provide a framework for future in vivo work. We previously demonstrated that with DSM265 alone, the in vitro data is a good predictor of the in vivo evolution of resistance (*Mandt et al., 2019*). It will be important to test potential combination strategies in vivo, as pharmacokinetics and pharmacodynamics may impact how multiple drugs interact with each other, with implications for the emergence of resistance. However, our continued identification of novel point mutations in the *dhodh* locus suggests that we still have not saturated the *dhodh* resistome. Additional mutational pathways to resistance could potentially be identified through further resistance selections or high-throughput methods such as deep mutational scanning to assess variants. The mutational flexibility of the inhibitor binding site in DHODH increases the likelihood of resistance to any compound or compound combination that targets this site. CNV also appears to be a general mechanism of resistance to multiple chemical classes of DHODH inhibitors. One implication of these results is that targeting this enzyme with antimalarial drugs has a high risk of resistance emergence and brings into question the usefulness of pursuing further DHODH inhibitors.

## Methods

### Reagents and parasite lines

Atovaquone, dihydroartemisinin, and mefloquine were purchased from Sigma-Aldrich (St. Louis, MO, USA). IDI-6273 was purchased from ChemDiv (San Diego, CA, USA). Genz669178 was graciously provided by Genzyme, a Sanofi Company (Waltham, MA, USA). DSM265 was a kind gift from Margaret Phillips of University of Texas Southwestern (Dallas, TX, USA). The *Pf*3D7 A10 clone used in this study was adapted to a 40 hr replication cycle in in vitro culture, and has been previously used in large-scale selection and sequencing efforts (*Cowell et al., 2018*). The identity of the parental parasite lines used in this study (3D7 A10 and DHODH C276Y) were verified by WGS and alignment to the reference genome. The parental parasite cultures tested negative for mycoplasma using the LookOut Mycoplasma PCR Detection Kit (Sigma). The identity of clones generated over the course of the study were also validated by sequencing and *Mycoplasma* status tested as above. All cell lines were negative for mycoplasma. Parasites with point mutations in *Pf*DHODH derived from previous work were generated as described and identity verified by sequencing (*Mandt et al., 2019*; *Ross et al., 2014*). Compounds from GSK libraries were previously published in *Ross et al., 2018* (Key Resources Table).

### In vitro parasite culturing

Parasites were cultured by standard methods with RPMI 1640 (Life Technologies) in 5% O$^+$ human blood obtained from Interstate Blood Bank. RPMI 1640 was supplemented with 28 mM NaHCO$_3$, 25 mM HEPES, 50 mg/mL hypoxanthine, 2525 µg/mL gentamycin, and 0.5% Albumax II (Life Technologies). Parasite cultures were maintained in a gas mixture of 95% N$_2$, 4% CO$_2$, and 1.1% O$_2$ at 37°C. Parasites were regularly synchronized by 5% sorbitol treatment (*Lambros and Vanderberg, 1979*), and all lines generated were preserved in glycerol stocks stored in liquid nitrogen.

### Resistance selections

In vitro resistance selections were performed in triplicate 25 mL flasks with 3–4% ring-stage parasitemia. Selected populations were treated with the indicated dose of compound in RPMI daily until parasites were no longer visible by thin smear microscopy. Cultures were then replenished with compound-free RPMI and monitored every 2–5 days by microscopy. Bulk recrudescent cultured were phenotyped in dose-response assays. Resistant populations were cloned by limiting dilution in 96-well plates at a density of 0.2 parasites per well in the absence of compound. For complete list of cell lines generated, see Key Resources Table.

### Dose-response assays

To assess susceptibility phenotypes of parasite lines, ring-stage parasites were grown at 1% hematocrit, 1% starting parasitemia in 40 µL volumes in 384-well black clear-bottomed plates. Parasites were cultured in the presence of test compounds plated in triplicate concentrations in 12-point serial dilutions, and parasite growth after 72 hr was measured by a SYBR Green-based fluorescence assay (*Johnson et al., 2007*; *Smilkstein et al., 2004*). Lysis buffer with SYBR Green I at 10× concentration was added to the plates and allowed to incubate at room temperature for at least 12 hr. Fluorescence

was measured at 494 nm excitation, 530 nm emission. Data was analyzed with CCD Vault, which calculates $EC_{50}$ values based on a non-linear dose-response curve fit (Burlingame, CA, USA).

## Genomic DNA analysis

For genetic characterization of resistant clones, red blood cells infected with late-stage parasites were washed with 0.1% saponin to obtain isolated parasite pellets. For competitive growth assays, ring-stage infected red blood cells were directly suspended in DNA/RNA Shield (Zymo Research). Genomic DNA was then extracted using a DNAeasy Blood and Tissue Kit (QIAGEN).

### Whole-genome sequencing

Libraries were prepared with the standard dual index protocol of the Nextera XT kit (Cat. No FC-131-1024). WGS was performed on an Illumina NovaSeq 6000 with S4 200 chemistry and 100 bp paired end reads. The *P. falciparum* 3D7 genome (PlasmoDB v. 13.0) was used as a reference for read alignment. Variants were called using an established analysis pipeline as previously described, where mutations are identified that are present in selected parasites that are not in the original parent line (*Cowell et al., 2018*). CNV was determined based on differential DNA sequence coverage as previously described (*Cowell et al., 2018*). A table with the coverage for each run is given in *Supplementary file 1g*.

### PCR and Sanger sequencing

Six of the ten clones from the DSM265 + TCMDC-125334 combination selection were genetically characterized by Sanger sequencing. PCR amplification of the *dhodh* locus was performed using Phusion High-Fidelity PCR Master Mix (New England BioLabs) as per the protocol. The locus was amplified in three overlapping segments with primers listed in the Key Resources Table.

### Quantitative PCR to assess CNV

Copy number of the *dhodh* locus was assessed by quantitative PCR (qPCR) as previously described (*Ross et al., 2014*), with primers listed in the Key Resources Table with power SYBR Green Master Mix (Applied Biosystems) and 0.1 ng gDNA template. qPCR was performed on a ViiA7 real-time PCR system with 384-well block (Applied Biosystems), and relative copy number was calculated using the $\Delta\Delta C_T$ method.

## In silico molecular modeling

For the modeling simulations conducted with Flare v.4.0.3.40719 (Cresset-Group), the DHODH crystal structures (PDB codes: 4*R*X0, 3O8A, 4CQ8, 4CQA, and 4CQ9) were imported from the PDB and 'Protein Prep' was performed using the default settings. Sequences were aligned and all structures superimposed to 4*R*X0. Three forms of TCMDC-125334 (unspecified enantiomer, R-isomer, and S-isomer) were then docked to all sequences using ensemble docking with the docking grid defined as all residues. Docking was performed using 'Very Accurate but Slow' settings modified to allow rotation about amide bonds. Mutant structures were prepared with Flare's mutate function using default settings, energy minimized with Flare's protein minimize function in the absence of other molecules and then in the presence of TCMDC-125334, orotate, and FMN, and then rescored as described above. Simulations were visualized using PyMOL v. 4.6.0 (Intel).

## Competition growth assays

Mutant and wildtype cultures were synchronized at least twice prior to the assay. Ring-stage mutant and wildtype parasites at 5% hematocrit and 1% parasitemia were combined at equal volumes. Mixed culture was split into three 10 mL replicate flasks, which were maintained for 4 weeks. Genomic DNA samples were collected and dose-response assays performed at indicated timepoints.

## Statistical analysis

The heatmap visualizing patterns of cross-resistance and collateral sensitivity was created using Multi-ExperimentViewer (MeV) version 4.9.0. Hierarchical clustering of both parasite lines and compounds was performed based on Euclidean distance using average linkage. Prism v8 (GraphPad) was used

to make graphs and to perform statistical analysis comparing $EC_{50}$'s of selected parasite clones and wildtype 3D7 A10. Significance was determined using a non-parametric ANOVA test (Kruskall-Wallis) with post hoc multiple comparisons (Dunn's test).

## Acknowledgements

We thank M Phillips (UT Southwestern) and J Burrows (Medicines for Malaria Venture) for generously providing compounds DSM265 and DSM267. We are also grateful to P Hinkson (Harvard TH Chan School of Public Health) for technical support. We are grateful for generous financial support by the NIH (grant no. R01 AI093716 to DFW), the Bill and Melinda Gates Foundation (Grand Challenges Exploration grant no. OPP1132451 to DFW, AKL, and FJG.), and the Harvard Malaria Initiative with support from ExxonMobil Foundation (to DFW). REKM was additionally supported by the Harvard Herchel Smith Fellowship, an NIH T32 grant, and funds from the ExxonMobil Foundation. MRL was supported in part by a Ruth L Kirschstein Institutional National Research Service Award T32 GM008666 from the National Institute of General Medical Sciences. This publication includes data generated at the UC San Diego IGM Genomics Center utilizing an Illumina NovaSeq 6000 that was purchased with funding from a National Institutes of Health SIG grant (#S10 OD026929). R.E.K.M. present address: National Institute of Allergy and Infectious Diseases, National Institutes of Health. Bethesda, Maryland. The work discussed in this article was performed by R.E.K.M. elsewhere prior to becoming a federal employee. The article was written/edited by R.E.K.M. in her private capacity. No official support or endorsement by the is intended or should be inferred.

## Additional information

### Competing interests

Rebecca EK Mandt: The work discussed in this article was performed by REKM elsewhere prior to becoming a federal employee. The article was written/edited by REKM in her private capacity. No official support or endorsement by the National Institute of Allergy and Infectious Diseases is intended or should be inferred. Elizabeth A Winzeler: Sits on the advisory board of the Tres Cantos Open Lab Foundation. Maria Jose Lafuente-Monasterio, Francisco Javier Gamo: GlaxoSmithKline employee. Dyann F Wirth: Sits on the advisory board of Medicines for Malaria Venture. The other authors declare that no competing interests exist.

### Funding

| Funder | Grant reference number | Author |
|---|---|---|
| National Institutes of Health | R01 AI093716 | Rebecca EK Mandt<br>Dyann F Wirth<br>Amanda K Lukens |
| Bill and Melinda Gates Foundation | OPP1132451 | Rebecca EK Mandt<br>Maria Jose Lafuente-Monasterio<br>Francisco Javier Gamo<br>Dyann F Wirth<br>Amanda K Lukens |
| National Institutes of Health | T32 GM008666 | Madeline R Luth |
| ExxonMobil Foundation | | Rebecca EK Mandt<br>Dyann F Wirth<br>Amanda K Lukens |
| National Institutes of Health | R01 AI169892 | Amanda K Lukens<br>Dyann F Wirth<br>Elizabeth A Winzeler |

The funders had no role in study design, data collection and interpretation, or the decision to submit the work for publication.

## Author contributions
Rebecca EK Mandt, Conceptualization, Data curation, Investigation, Visualization, Methodology, Writing – original draft; Madeline R Luth, Data curation, Software, Methodology, Writing – original draft, Writing - review and editing; Mark A Tye, Software, Investigation, Writing – original draft, Writing - review and editing; Ralph Mazitschek, Supervision, Writing - review and editing; Sabine Ottilie, Supervision, Project administration, Writing - review and editing; Elizabeth A Winzeler, Dyann F Wirth, Conceptualization, Supervision, Writing - review and editing; Maria Jose Lafuente-Monasterio, Francisco Javier Gamo, Conceptualization, Writing - review and editing; Amanda K Lukens, Conceptualization, Supervision, Project administration, Writing - review and editing

## Author ORCIDs
Rebecca EK Mandt ⬤ https://orcid.org/0000-0001-7165-7876
Elizabeth A Winzeler ⬤ http://orcid.org/0000-0002-4049-2113
Amanda K Lukens ⬤ http://orcid.org/0000-0002-9560-7643

## Decision letter and Author response
Decision letter https://doi.org/10.7554/eLife.85023.sa1
Author response https://doi.org/10.7554/eLife.85023.sa2

# Additional files

## Supplementary files
• Supplementary file 1. Supplementary tables with additional experimental information for 'Diverse evolutionary pathways pose a challenge to the use of collateral sensitivity as a strategy to suppress resistance'.

• MDAR checklist

## Data availability
The raw whole-genome sequencing data generated in this study have been submitted to the NCBI Sequence Read Archive database (https://www.ncbi.nlm.nih.gov/sra/) under accession number PRJNA689594. Sanger sequencing of the PCR amplified dhodh locus have been submitted to GenBank (NCBI) under accession numbers MZ571149-MZ571158.

The following datasets were generated:

| Author(s) | Year | Dataset title | Dataset URL | Database and Identifier |
|---|---|---|---|---|
| Mandt REK, Luth MR, Tye MA, Mazitschek R, Ottilie S, Winzeler EA, Lafuente-Monasterio MJ, Gamo FJ, Wirth DF, Lukens AK | 2022 | *Plasmodium falciparum* strain 3D7 A10 clone DT-F1-C1 dihydroorotate dehydrogenase (dhodh) gene, complete cds | https://www.ncbi.nlm.nih.gov/nuccore/MZ571149 | NCBI Nucleotide, MZ571149 |
| Mandt REK, Luth MR, Tye MA, Mazitschek R, Ottilie S, Winzeler EA, Lafuente-Monasterio MJ, Gamo FJ, Wirth DF, Lukens AK | 2022 | *Plasmodium falciparum* strain 3D7 A10 clone DT-F1-C2 dihydroorotate dehydrogenase (dhodh) gene, complete cds | https://www.ncbi.nlm.nih.gov/nuccore/MZ571150 | NCBI Nucleotide, MZ571150 |
| Mandt REK, Luth MR, Tye MA, Mazitschek R, Ottilie S, Winzeler EA, Lafuente-Monasterio MJ, Gamo FJ, Wirth DF, Lukens AK | 2022 | *Plasmodium falciparum* strain 3D7 A10 clone DT-F1-C3 dihydroorotate dehydrogenase (dhodh) gene, complete cds | https://www.ncbi.nlm.nih.gov/nuccore/MZ571151 | NCBI Nucleotide, MZ571151 |

*Continued*

| Author(s) | Year | Dataset title | Dataset URL | Database and Identifier |
|---|---|---|---|---|
| Mandt REK, Luth MR, Tye MA, Mazitschek R, Ottilie S, Winzeler EA, Lafuente-Monasterio MJ, Gamo FJ, Wirth DF, Lukens AK | 2022 | *Plasmodium falciparum* strain 3D7 A10 clone DT-F1-C4 dihydroorotate dehydrogenase (dhodh) gene, complete cds | https://www.ncbi.nlm.nih.gov/nuccore/MZ571152 | NCBI Nucleotide, MZ571152 |
| Mandt REK, Luth MR, Tye MA, Mazitschek R, Ottilie S, Winzeler EA, Lafuente-Monasterio MJ, Gamo FJ, Wirth DF, Lukens AK | 2022 | *Plasmodium falciparum* strain 3D7 A10 clone DT-F2-C1 dihydroorotate dehydrogenase (dhodh) gene, complete cds | https://www.ncbi.nlm.nih.gov/nuccore/MZ571153 | NCBI Nucleotide, MZ571153 |
| Mandt REK, Luth MR, Tye MA, Mazitschek R, Ottilie S, Winzeler EA, Lafuente-Monasterio MJ, Gamo FJ, Wirth DF, Lukens AK | 2022 | *Plasmodium falciparum* strain 3D7 A10 clone DT-F2-C2 dihydroorotate dehydrogenase (dhodh) gene, complete cds | https://www.ncbi.nlm.nih.gov/nuccore/MZ571154 | NCBI Nucleotide, MZ571154 |
| Mandt REK, Luth MR, Tye MA, Mazitschek R, Ottilie S, Winzeler EA, Lafuente-Monasterio MJ, Gamo FJ, Wirth DF, Lukens AK | 2022 | *Plasmodium falciparum* strain 3D7 A10 clone DT-F2-C3 dihydroorotate dehydrogenase (dhodh) gene, complete cds | https://www.ncbi.nlm.nih.gov/nuccore/MZ571155 | NCBI Nucleotide, MZ571155 |
| Mandt REK, Luth MR, Tye MA, Mazitschek R, Ottilie S, Winzeler EA, Lafuente-Monasterio MJ, Gamo FJ, Wirth DF, Lukens AK | 2022 | *Plasmodium falciparum* strain 3D7 A10 clone DT-F2-C4 dihydroorotate dehydrogenase (dhodh) gene, complete cds | https://www.ncbi.nlm.nih.gov/nuccore/MZ571156 | NCBI Nucleotide, MZ571156 |
| Mandt REK, Luth MR, Tye MA, Mazitschek R, Ottilie S, Winzeler EA, Lafuente-Monasterio MJ, Gamo FJ, Wirth DF, Lukens AK | 2022 | *Plasmodium falciparum* strain 3D7 A10 clone DT-F2-C5 dihydroorotate dehydrogenase (dhodh) gene, complete cds | https://www.ncbi.nlm.nih.gov/nuccore/MZ571157 | NCBI Nucleotide, MZ571157 |
| Mandt REK, Luth MR, Tye MA, Mazitschek R, Ottilie S, Winzeler EA, Lafuente-Monasterio MJ, Gamo FJ, Wirth DF, Lukens AK | 2022 | *Plasmodium falciparum* strain 3D7 A10 clone DT-F2-C6 dihydroorotate dehydrogenase (dhodh) gene, complete cds | https://www.ncbi.nlm.nih.gov/nuccore/MZ571158 | NCBI Nucleotide, MZ571158 |
| University of California, San Diego | 2021 | WGS of *Plasmodium falciparum* selected with DHODH inhibitors | https://www.ncbi.nlm.nih.gov/bioproject/PRJNA689594/ | NCBI BioProject, PRJNA689594 |

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

# Appendix 1

## Appendix 1—key resources table

| Reagent type (species) or resource | Designation | Source or reference | Identifiers | Additional information |
|---|---|---|---|---|
| Gene (*Plasmodium falciparum*) | *dhodh; Pfdhodh* | PlasmoDB | PF3D7_0603300 | |
| Strain, strain background (*Plasmodium falciparum*) | 3D7 A10 | Goldberg lab at Washington University, St. Louis, MO, USA | 3D7 A10; 3D7_A10 | *Cowell et al., 2018* |
| Cell line (*Plasmodium falciparum*) | T-F1-C1 | This paper | | Isolated from selection of 3D7 A10 parasites with TCMDC-125334; available upon request |
| Cell line (*Plasmodium falciparum*) | T-F1-C2 | This paper | | Isolated from selection of 3D7 A10 parasites with TCMDC-125334; available upon request |
| Cell line (*Plasmodium falciparum*) | T-F1-C3 | This paper | | Isolated from selection of 3D7 A10 parasites with TCMDC-125334; available upon request |
| Cell line (*Plasmodium falciparum*) | T-F2-C1 | This paper | | Isolated from selection of 3D7 A10 parasites with TCMDC-125334; available upon request |
| Cell line (*Plasmodium falciparum*) | T-F2-C2 | This paper | | Isolated from selection of 3D7 A10 parasites with TCMDC-125334; available upon request |
| Cell line (*Plasmodium falciparum*) | T-F2-C3 | This paper | | Isolated from selection of 3D7 A10 parasites with TCMDC-125334; available upon request |
| Cell line (*Plasmodium falciparum*) | T-F2-C4 | This paper | | Isolated from selection of 3D7 A10 parasites with TCMDC-125334; available upon request |
| Cell line (*Plasmodium falciparum*) | T-F2-C5 | This paper | | Isolated from selection of 3D7 A10 parasites with TCMDC-125334; available upon request |
| Cell line (*Plasmodium falciparum*) | C276Y Parent | *Mandt et al., 2019* | S1-F1-C1 | |
| Cell line (*Plasmodium falciparum*) | D-T-F1-C1 | This paper | | Isolated from selection of C276Y parent parasites with TCMDC-125334; available upon request |
| Cell line (*Plasmodium falciparum*) | D-T-F1-C2 | This paper | | Isolated from selection of C276Y parent parasites with TCMDC-125334; available upon request |
| Cell line (*Plasmodium falciparum*) | D-T-F1-C3 | This paper | | Isolated from selection of C276Y parent parasites with TCMDC-125334; available upon request |
| Cell line (*Plasmodium falciparum*) | D-T-F1-C4 | This paper | | Isolated from selection of C276Y parent parasites with TCMDC-125334; available upon request |
| Cell line (*Plasmodium falciparum*) | D-T-F2-C1 | This paper | | Isolated from selection of C276Y parent parasites with TCMDC-125334; available upon request |
| Cell line (*Plasmodium falciparum*) | D-T-F2-C2 | This paper | | Isolated from selection of C276Y parent parasites with TCMDC-125334; available upon request |
| Cell line (*Plasmodium falciparum*) | D-T-F2-C3 | This paper | | Isolated from selection of C276Y parent parasites with TCMDC-125334; available upon request |

*Appendix 1 Continued on next page*

*Appendix 1 Continued*

| Reagent type (species) or resource | Designation | Source or reference | Identifiers | Additional information |
|---|---|---|---|---|
| Cell line (*Plasmodium falciparum*) | D-T-F2-C4 | This paper | | Isolated from selection of C276Y parent parasites with TCMDC-125334; available upon request |
| Cell line (*Plasmodium falciparum*) | D-T-F3-C1 | This paper | | Isolated from selection of C276Y parent parasites with TCMDC-125334; available upon request |
| Cell line (*Plasmodium falciparum*) | D-T-F3-C2 | This paper | | Isolated from selection of C276Y parent parasites with TCMDC-125334; available upon request |
| Cell line (*Plasmodium falciparum*) | D-T-F3-C3 | This paper | | Isolated from selection of C276Y parent parasites with TCMDC-125334; available upon request |
| Cell line (*Plasmodium falciparum*) | DT-F1-C1 | This paper | | Isolated from selection of 3D7 A10 parasites with DSM265 and TCMDC-125334; available upon request |
| Cell line (*Plasmodium falciparum*) | DT-F1-C2 | This paper | | Isolated from selection of 3D7 A10 parasites with DSM265 and TCMDC-125334; available upon request |
| Cell line (*Plasmodium falciparum*) | DT-F1-C3 | This paper | | Isolated from selection of 3D7 A10 parasites with DSM265 and TCMDC-125334; available upon request |
| Cell line (*Plasmodium falciparum*) | DT-F1-C4 | This paper | | Isolated from selection of 3D7 A10 parasites with DSM265 and TCMDC-125334; available upon request |
| Cell line (*Plasmodium falciparum*) | DT-F2-C1 | This paper | | Isolated from selection of 3D7 A10 parasites with DSM265 and TCMDC-125334; available upon request |
| Cell line (*Plasmodium falciparum*) | DT-F2-C2 | This paper | | Isolated from selection of 3D7 A10 parasites with DSM265 and TCMDC-125334; available upon request |
| Cell line (*Plasmodium falciparum*) | DT-F2-C3 | This paper | | Isolated from selection of 3D7 A10 parasites with DSM265 and TCMDC-125334; available upon request |
| Cell line (*Plasmodium falciparum*) | DT-F2-C4 | This paper | | Isolated from selection of 3D7 A10 parasites with DSM265 and TCMDC-125334; available upon request |
| Cell line (*Plasmodium falciparum*) | DT-F2-C5 | This paper | | Isolated from selection of 3D7 A10 parasites with DSM265 and TCMDC-125334; available upon request |
| Cell line (*Plasmodium falciparum*) | DT-F2-C6 | This paper | | Isolated from selection of 3D7 A10 parasites with DSM265 and TCMDC-125334; available upon request |
| Sequence-based reagent | DHODH F1A | *Mandt et al., 2019* | PCR primer | GTGTGATAGATAGCTCCAGTCG |
| Sequence-based reagent | DHODH R1B | *Mandt et al., 2019* | PCR primer | CGTTTGGCCCCTTGGGGTTATGG |
| Sequence-based reagent | DHODH F2A | *Mandt et al., 2019* | PCR primer | TTGATGGTGAAATATGTCATGACCTT |
| Sequence-based reagent | DHODH R2A | *Mandt et al., 2019* | PCR primer | CCAAGGGCTTCTTTTTTGTTGTATTAAACC |
| Sequence-based reagent | DHODH F3A | *Mandt et al., 2019* | PCR primer | GTCACATGATGAAAGATGCTAAGG |
| Sequence-based reagent | DHODH R3B | *Mandt et al., 2019* | PCR primer | CGCACTTATGTGTCGCCCG |

*Appendix 1 Continued on next page*

*Appendix 1 Continued*

| Reagent type (species) or resource | Designation | Source or reference | Identifiers | Additional information |
|---|---|---|---|---|
| Sequence-based reagent | DHODH Front-F | *Guler et al., 2013* | qPCR primer | TCCATTCGGTGTTGCTGCAGGATTTGAT |
| Sequence-based reagent | DHODH Front-R | *Guler et al., 2013* | qPCR primer | TCTGTAACTTTGTCACAACCCATATTA |
| Sequence-based reagent | DHODH Rear-F | *Guler et al., 2013* | qPCR primer | GTGTTAGCGGAGCAAAACTAAAAG |
| Sequence-based reagent | DHODH Rear-R | *Guler et al., 2013* | qPCR primer | ATAATTGACAAACTGAAGCACCTG |
| Sequence-based reagent | Seryl t-RNA Synthetase-F | *Guler et al., 2013* | qPCR primer | GGAACAATTCTGTATTGCTTTACC |
| Sequence-based reagent | Seryl t-RNA Synthetase-R | *Guler et al., 2013* | qPCR primer | AAGCTGCGTTGTTTAAAGCTC |
| Sequence-based reagent | 18s Ribosomal RNA | *Guler et al., 2013* | qPCR primer | ACAATTCATCATATCTTTCAATCGGTA |
| Sequence-based reagent | 18s Ribosomal RNA | *Guler et al., 2013* | qPCR primer | GCTGACTACGTCCCTGCCC |
| Commercial assay or kit | Nextera XT DNA Library Preparation Kit | Illumina | Cat # FC-131–1024 | |
| Commercial assay or kit | DNeasy Blood & Tissue Kit | QIAGEN | Cat # 69556 | |
| Commercial assay or kit | LookOut Mycoplasma PCR Detection Kit | Sigma | Cat # MP0035 | |
| Commercial assay or kit | Phusion High-Fidelity PCR Master Mix | New England BioLabs | Cat No: M0531S | |
| Commercial assay or kit | DNA/RNA Shield | Zymo Research | Cat No: R1200-25 | |
| Commercial assay or kit | SYBR Green Master Mix | Applied Biosystems | Cat No: A46109 | |
| Chemical compound, drug | IDI-6273 | ChemDiv | ChemDiv4861-0080 | |
| Chemical compound, drug | Genz669178 | Genzyme, a Sanofi Company | | |
| Chemical compound, drug | DSM265 | Laboratory of Margaret Phillips, University of Texas Southwestern, Dallas, TX, USA | | *Phillips et al., 2015* |
| Chemical compound, drug | Atovaquone | Sigma-Aldrich | CAS No: 95233-18-4; Product No: A7986 | |
| Chemical compound, drug | Dihydroartemisinin | Sigma-Aldrich | CAS No: 71939-50-9; Product No: 1200520 | |
| Chemical compound, drug | TCMDC-125334 | *Ross et al., 2018* | Additional material ordered from MolPort: MolPort-004-150-355 | |
| Software, algorithm | HaplotypeCaller | Genome Analysis Toolkit (GATK) | | See also: *Cowell et al., 2018* |
| Software, algorithm | Platypus pipeline | https://sourceforge.net/projects/platypusmga/ ; *Manary et al., 2014* | | See also: *Cowell et al., 2018* |
| Software, algorithm | *Plasmodium* CNV analysis pipeline | rwillia2001 /plasmodium_cnv_analysis — Bitbucket | | *Cowell et al., 2018* |
| Software, algorithm | Prism v9 | GraphPad | | |
| Software, algorithm | CDD Vault | Collaborative Drug Discovery Inc. | | |

*Appendix 1 Continued*

| Reagent type (species) or resource | Designation | Source or reference | Identifiers | Additional information |
|---|---|---|---|---|
| Software, algorithm | MultiExperimentViewer (MeV) version 4.9.0 | https://sourceforge.net/projects/mev-tm4/ | | |
| Software, algorithm | Applied Biosystems QuantStudio Real-Time PCR Software | Thermo Fisher Scientific | | |
| Software, algorithm | Flare v.4.0.3.40719 | Cresset-GroupTM | | |
| Software, algorithm | PyMOL v. 4.6.0 | Intel | | |

