## [Editor Report]

This study addresses an important question in the field of antimicrobial chemotherapy: whether combinations of enzyme inhibitors that select for mutations that confer resistance to one inhibitor and at the same time increased sensitization to the other inhibitor, can provide a path towards mitigating resistance risks. The authors investigated one such combination of inhibitors of *Plasmodium falciparum* DHODH (dihydroorotate dehydrogenase), finding that despite "collateral sensitivity", it was still possible to select parasites with resistance to both inhibitors without any change in parasite fitness. Additional cross-susceptibility and structural modelling strengthen this study, which is performed to a high technical standard and presents a convincing body of data.

---

## [Decision Letter]

**Decision letter after peer review:**

Thank you for submitting your article "Collateral sensitivity as a strategy to suppress resistance emergence: the challenge of diverse evolutionary pathways" for consideration by *eLife*. Your article has been reviewed by 3 peer reviewers, including Christine Clayton as Reviewing Editor and Reviewer #1, and the evaluation has been overseen by Dominique Soldati-Favre as the Senior Editor.

Essential revisions:

a) In summarizing and in the discussion, please make it clear that the detailed results concern only one pair of inhibitors. Remove unsupported broad generalizations (or else, say that they are possible interpretations but would need to be confirmed with additional compound combinations). Mention copy-number variation in the Abstract.

b) Figure 3 panel E – please provide EC50s also for the measurements currently labelled as ">", as requested by reviewer 2.

c) The variation in font sizes in the Figures is unacceptable. Please use similar font sizes for all labels in all Figures. Everything must be easy to read at 100% on a laptop. Also please implement other suggestions concerning the intelligibility of the Figures, such as colour choices.

d) In tables, reduce the number of significant figures to reflect the accuracy of the data.

e) In all Figures/tables, describe what the statistical test was, how many replicates there were, and how they were obtained (how many independent experiments).

f) To completely understand the development of double resistance (and particularly, how serious a problem it would be) it would be necessary to measure the frequency of double resistance emergence compared with using either compound alone. Please state this clearly as a caveat to your results. You could also, for the 10^8^ inoculum, explicitly compare the times to obtain resistance both compounds and to each individual compound.

*Reviewer #1 (Recommendations for the authors):*

Sweeping general statements, such as "if a resistant mutant is competitively fit, treatment with a second drug will more likely result in the acquisition of additional genetic changes and subsequent multi-drug resistance" – while plausible – are not supported by the data. To make such a statement it would be necessary to test multiple different resistance mutations, and several different inhibitors and targets.

The combination of the two drugs resulted in a delay in the generation of resistance, but no quantitation is provided. Also, different selection protocols were used in the various experiments. What is the frequency of resistance emergence compared with using either compound alone? This is important.

*Reviewer #2 (Recommendations for the authors):*

Below are a number of suggestions for ways to improve the manuscript. Most relate to data presentation or the text. There is also one request to repeat some drug assays with DSM265 shown in Figure 3E to ensure that the IC50 values are accurately captured.

– In the Introduction, the authors should provide a broader overview of the function of DHODH. Later, when discussing their selection studies (typically conducted with ~10^8^ parasites) they should discuss how this aligns with prior studies that have selected for resistance and reported the minimum inoculum of resistance (MIR) values for this target (stated as log10 = 5.5 in PMID 34001441).

– Page 4 para 2: References 36-41 do not all refer to PfCRT. I would recommend removing 36 and 41 and replacing them with PMID 35130315

– Please spell out the abbreviation FMN (flavin mononucleotide).

– Figure 1 is an excellent way of presenting how mutations impact compound potency in terms of cross-resistance or collateral susceptibility. Panel B should relabel the text on the right, which is currently in poor resolution. The legend should also clarify – was this always the same parental line for each mutant? Panel C should indicate this is a log10 scale (on the panel and also the legend). Minor ticks should be used to add more resolution to the Y axes (e.g. placed between the 0 and log10 values of 2 or -2, which equates to 100-fold higher or lower values). Also, given that panels B and C are presumably the same parasite data, why does it extend to log10 values of 2 or -2 in C and only 1 or -1 in B? Please clarify.

– Starting on page 7, the authors state: "Of the clones isolated from flask 2, four did not have a resistance phenotype, and had no genetic changes in the dhodh locus (Figure 2C-F, Table 1, Figure S2)". The authors should indicate that Figure 2 illustrates only one clone, F2-C2, with no CNV or SNP in DHODH and no change in IC50. There is only one other clone shown, which is F2-C1 which has an I263S mutation. Do the authors ascribe this to the removal of drug pressure during the expansion of these clones?

– For Table 1, I would recommend reducing the number of significant digits to 1 for values below 10 nM (for example, writing 5.1 nM, not 5.07 nM). This would be more in line with the accuracy of these data. The same applies (less often) to Tables 2-4 and Figure S3B (note: there is no "A" listed in Figure S3).

– Figure 3: Panel E has highly unconventional labeling with ">values shown". The legend indicates that "In cases where the EC50 could not be determined with a range of concentrations tested, the maximum concentration that achieved >50% growth was used as a stand-in value.". These are intermediate values that should be entirely feasible to measure. Rather than setting this new standard, the authors should repeat enough assays to provide accurate mean {plus minus} SD values. Also, the authors should review each panel and add sufficient minor ticks to allow readers to estimate values. Panel G also has a labeling error – the green dose-response line should be C276Y/F227Y (not C276Y as is also shown for the blue line). Also, the legend needs improvement. in vitro dose needs italics only for in vitro and not for dose. The legend also states "th DHODH C276Y/F227Y". The legend also states that the F227Y mutant was selected using DSM265, yet the next textual paragraph below shows that this mutant was selected using DSM267. Which is it?

– Figure 3 (more): This legend also shows that the dose-response data showed SDs of technical replicates. The legend should clarify whether this means independent assays or technical replicates of a single assay. Data File S1 shows that these are mostly three independent bio-reps with 3D7-A10 having 12-13. The legend should refer to this Data file, as should the legends of all Figures that show data outlined in that file.

– The qPCR data refer to DHODH front or DHODH rear. Please define these terms.

– Figure 4C: The X axis should read DSM265 not DSM-265.

– Figure S2 legend: Please specify the type of error bar and numbers of bio-reps and technical replicates. Also, please specify what is meant by DHODH front vs rear. Finally, it would likely be better to tailor each Y axis to the data sets shown in each panel, rather than having several panels show the data only represented at the bottom of the graph. Please be sure to place enough tick marks to allow the reader to more easily estimate relative copy numbers. Also, please explain in the legend why Flask 2 from Round 1 was not used in Round 2.

– Page 17: Please correct "by the calculating".

*Reviewer #3 (Recommendations for the authors):*

1. Since the study "brings into question the usefulness of pursuing further DHODH inhibitors", I wonder if the authors could develop their thinking further on this important point.

a. Would deep mutational scanning be useful here, for example?

b. Is the view here that those previously tested combinations would indeed also fail if pressured further for cross-resistance?

c. Do the authors believe that CNV could be a general mechanism that would undermine other collateral resistance strategies?

d. Is the broader conclusion here that it would be advisable to combine drugs against two (or more) independent targets rather than using two drugs against one target?

2. Could the authors clarify the relationship between the compound screen reported in 2018 (ref 47) and the compound screen reported in Figure 1B here? Is there any overlap? Compound 17 looks to have been a promising potential candidate for combination follow-up in the prior study.

3. I'm assuming that no 'non-DHODH' mutations were detected by whole-genome sequencing, but I'm not sure if this is stated in the manuscript. This is important as the genotype, and resistance phenotype relationships are not validated by gene editing, for example.

---

## [Author Response]

Essential revisions:a) In summarizing and in the discussion, please make it clear that the detailed results concern only one pair of inhibitors. Remove unsupported broad generalizations (or else, say that they are possible interpretations but would need to be confirmed with additional compound combinations). Mention copy-number variation in the Abstract.

We have revised the Discussion section to focus on one pair of inhibitors. We have also noted in the abstract that copy-number variation was a common mechanism of resistance to TCMDC-125334.

b) Figure 3 panel E – please provide EC50s also for the measurements currently labelled as ">", as requested by reviewer 2.

We have conducted additional drug assays with adjusted dose ranges that have allowed us to accurately calculate EC­_50_ values for Figure 3.

c) The variation in font sizes in the Figures is unacceptable. Please use similar font sizes for all labels in all Figures. Everything must be easy to read at 100% on a laptop. Also please implement other suggestions concerning the intelligibility of the Figures, such as colour choices.

We have increased and standardized the font size across all figures and made other requested changes to the figures.

d) In tables, reduce the number of significant figures to reflect the accuracy of the data.

We have reduced the number of significant figures in the tables based on Reviewer #2’s recommendations.

e) In all Figures/tables, describe what the statistical test was, how many replicates there were, and how they were obtained (how many independent experiments).

We have included additional information in the figure legends to describe the statistical test and replicates.

f) To completely understand the development of double resistance (and particularly, how serious a problem it would be) it would be necessary to measure the frequency of double resistance emergence compared with using either compound alone. Please state this clearly as a caveat to your results. You could also, for the 10^8^ inoculum, explicitly compare the times to obtain resistance both compounds and to each individual compound.

We have noted this in the Discussion section. See paragraph starting with: “We cannot quantify the frequency of resistance emergence based on our results…”

Reviewer #1 (Recommendations for the authors):Sweeping general statements, such as "if a resistant mutant is competitively fit, treatment with a second drug will more likely result in the acquisition of additional genetic changes and subsequent multi-drug resistance" – while plausible – are not supported by the data. To make such a statement it would be necessary to test multiple different resistance mutations, and several different inhibitors and targets.

We have edited the Discussion section per the reviewer’s suggestion.

The combination of the two drugs resulted in a delay in the generation of resistance, but no quantitation is provided. Also, different selection protocols were used in the various experiments. What is the frequency of resistance emergence compared with using either compound alone? This is important.

We do quantify the days to resistance emergence in the figures. This comparison is now described directly in the text. We also note the number of flasks in which resistance parasites emerge, but given that we are only using three populations per selection experiment, we cannot draw a conclusion about the frequency of resistance emergence based on our results. Doing so would require a Luria-Delbruck type experimental design that is beyond the scope of this study. We have provided information on the hematocrit, the starting inoculates, and the time to resistance.

Reviewer #2 (Recommendations for the authors):Below are a number of suggestions for ways to improve the manuscript. Most relate to data presentation or the text. There is also one request to repeat some drug assays with DSM265 shown in Figure 3E to ensure that the IC50 values are accurately captured.– In the Introduction, the authors should provide a broader overview of the function of DHODH. Later, when discussing their selection studies (typically conducted with ~10^8^ parasites) they should discuss how this aligns with prior studies that have selected for resistance and reported the minimum inoculum of resistance (MIR) values for this target (stated as log10 = 5.5 in PMID 34001441).

We have added additional context about DHODH to the introduction. We also noted in the Results section that the MIR for DSM265 is 10^5.5^.

– Page 4 para 2: References 36-41 do not all refer to PfCRT. I would recommend removing 36 and 41 and replacing them with PMID 35130315

We have added the indicated reference.

– Please spell out the abbreviation FMN (flavin mononucleotide).

We have spelled out the first use of the abbreviation FMN.

– Figure 1 is an excellent way of presenting how mutations impact compound potency in terms of cross-resistance or collateral susceptibility. Panel B should relabel the text on the right, which is currently in poor resolution. The legend should also clarify – was this always the same parental line for each mutant? Panel C should indicate this is a log10 scale (on the panel and also the legend). Minor ticks should be used to add more resolution to the Y axes (e.g. placed between the 0 and log10 values of 2 or -2, which equates to 100-fold higher or lower values). Also, given that panels B and C are presumably the same parasite data, why does it extend to log10 values of 2 or -2 in C and only 1 or -1 in B? Please clarify.

We have relabeled the text in Panel B and added information about the parental lines for each mutant. We have also indicated the axis is log10 scale, and added minor ticks to the Y-axes for panel C. We appreciate the reviewer’s observation about the range of the data in Panel B vs. Panel C. We have changed the scale in the heatmap in Panel B to log10=-2 to log10=2.

– Starting on page 7, the authors state: "Of the clones isolated from flask 2, four did not have a resistance phenotype, and had no genetic changes in the dhodh locus (Figure 2C-F, Table 1, Figure S2)". The authors should indicate that Figure 2 illustrates only one clone, F2-C2, with no CNV or SNP in DHODH and no change in IC50.

We note in the figure legend that the figure only shows two representative clones from each flask. The complete set of clones is described in Table 1. We have added an additional sentence to the text for each figure clarifying this.

There is only one other clone shown, which is F2-C1 which has an I263S mutation. Do the authors ascribe this to the removal of drug pressure during the expansion of these clones?

This is an interesting question. Our interpretation is that in the bulk culture there were both mutant and wildtype parasites. We did not examine the fitness of the DHODH I263S mutant line in the absence of drug.

– For Table 1, I would recommend reducing the number of significant digits to 1 for values below 10 nM (for example, writing 5.1 nM, not 5.07 nM). This would be more in line with the accuracy of these data. The same applies (less often) to Tables 2-4 and Figure S3B (note: there is no "A" listed in Figure S3).

We have adjusted the significant figures of the data per the reviewer’s recommendations.

– Figure 3: Panel E has highly unconventional labeling with ">values shown". The legend indicates that "In cases where the EC50 could not be determined with a range of concentrations tested, the maximum concentration that achieved >50% growth was used as a stand-in value.". These are intermediate values that should be entirely feasible to measure. Rather than setting this new standard, the authors should repeat enough assays to provide accurate mean {plus minus} SD values. Also, the authors should review each panel and add sufficient minor ticks to allow readers to estimate values. Panel G also has a labeling error – the green dose-response line should be C276Y/F227Y (not C276Y as is also shown for the blue line). Also, the legend needs improvement. in vitro dose needs italics only for in vitro and not for dose. The legend also states "th DHODH C276Y/F227Y". The legend also states that the F227Y mutant was selected using DSM265, yet the next textual paragraph below shows that this mutant was selected using DSM267. Which is it?

We have conducted additional drug assays with adjusted dose ranges that have allowed us to accurately calculate EC­_50_ values for Figure 3. We have also made the corrections to the figure labels and legends.

– Figure 3 (more): This legend also shows that the dose-response data showed SDs of technical replicates. The legend should clarify whether this means independent assays or technical replicates of a single assay. Data File S1 shows that these are mostly three independent bio-reps with 3D7-A10 having 12-13. The legend should refer to this Data file, as should the legends of all Figures that show data outlined in that file.

We have clarified that these are technical replicates of a single representative dose response assay. We also now refer to the tables and supplemental data files in the figure legends.

– The qPCR data refer to DHODH front or DHODH rear. Please define these terms.

We have clarified that DHODH front and DHODH rear refer to two amplicons within the DHODH gene.

– Figure 4C: The X axis should read DSM265 not DSM-265.

We have corrected the axis.

– Figure S2 legend: Please specify the type of error bar and numbers of bio-reps and technical replicates. Also, please specify what is meant by DHODH front vs rear. Finally, it would likely be better to tailor each Y axis to the data sets shown in each panel, rather than having several panels show the data only represented at the bottom of the graph. Please be sure to place enough tick marks to allow the reader to more easily estimate relative copy numbers. Also, please explain in the legend why Flask 2 from Round 1 was not used in Round 2.

We have updated the legend to note that DHODH front and DHODH rear refer to two amplicons within the dhodh gene. We have also added information about the technical replicates and error bars. We also adjusted the axes in the figures per the reviewer’s recommendation.

– Page 17: Please correct "by the calculating".

We have fixed this typo.

Reviewer #3 (Recommendations for the authors):1. Since the study "brings into question the usefulness of pursuing further DHODH inhibitors", I wonder if the authors could develop their thinking further on this important point.a. Would deep mutational scanning be useful here, for example?b. Is the view here that those previously tested combinations would indeed also fail if pressured further for cross-resistance?c. Do the authors believe that CNV could be a general mechanism that would undermine other collateral resistance strategies?d. Is the broader conclusion here that it would be advisable to combine drugs against two (or more) independent targets rather than using two drugs against one target?

We have expanded on this thinking in the discussion.

2. Could the authors clarify the relationship between the compound screen reported in 2018 (ref 47) and the compound screen reported in Figure 1B here? Is there any overlap? Compound 17 looks to have been a promising potential candidate for combination follow-up in the prior study.

We have clarified this relationship when discussing the results for Figure 1B. Compound 17 from the 2018 high-throughput screen is TCMDC-125334.

3. I'm assuming that no 'non-DHODH' mutations were detected by whole-genome sequencing, but I'm not sure if this is stated in the manuscript. This is important as the genotype, and resistance phenotype relationships are not validated by gene editing, for example.

There were additional mutations outside the dhodh locus identified via whole-genome sequencing. Based on the mutation rate and time in culture we expect a certain amount of variation to occur (see PMID:23408914). However, we found no instances of variation outside of DHODH appearing in multiple independent selections. We have noted this in the Discussion section. We are also making the whole genome sequence data publicly available.